# Continuous transcription initiation guarantees robust repair of all transcribed genes and regulatory regions

Anastasios Liakos[1,2,4], Dimitris Konstantopoulos [1,3,4], Matthieu D. Lavigne [1✉] & Maria Fousteri [1✉]

Inhibition of transcription caused by DNA damage-impaired RNA polymerase II (Pol II) elongation conceals a local increase in de novo transcription, slowly progressing from Transcription Start Sites (TSSs) to gene ends. Although associated with accelerated repair of Pol II-encountered lesions and limited mutagenesis, it is still unclear how this mechanism is maintained during genotoxic stress-recovery. Here we uncover a widespread gain in chromatin accessibility and preservation of the active H3K27ac mark after UV-irradiation. The concomitant increase in Pol II escape from promoter-proximal pause (PPP) sites of most active genes, PROMPTs and enhancer RNAs favors unrestrained initiation, as evidenced by the synthesis of nascent RNAs including start RNAs. Accordingly, drug-inhibition of PPP-release replenishes levels of pre-initiating Pol II at TSSs after UV. Our data show that such continuous engagement of Pol II molecules ensures maximal transcription-driven repair throughout expressed genes and regulatory loci. Importantly, revealing this unanticipated regulatory layer of UV-response provides physiological relevant traction to the emerging concept that Pol II initiation rate is determined by pause-release dynamics.

[1] Institute for Fundamental Biomedical Research, BSRC 'Alexander Fleming', 34 Fleming st., 16672 Vari, Athens, Greece. [2] Department of Biology, School of Science, National & Kapodistrian University of Athens, Athens, Greece. [3] Department of Biology, University of Crete, 70013 Herakleion, Greece. [4] These authors contributed equally: Anastasios Liakos, Dimitris Konstantopoulos. ✉email: lavigne@fleming.gr; fousteri@fleming.gr

Pol II initiation at TSSs and release into productive elongation from PPP are ubiquitous and crucial steps regulating transcription of protein-coding genes and long non-coding RNAs[1,2] (together called mRNAs in this paper). Similarly, Pol II is regulated for transcribing regulatory non-coding regions: enhancer RNAs (eRNAs) are expressed bidirectionally from eTSSs[3–5], while PROMoter uPstream transcripts (or upstream antisense RNAs, collectively called PROMPTs here) are produced in the opposite direction to mRNA when two stable transcripts are not initiated in very close proximity and in opposite directions (bidirectional promoters)[6]. However, contrary to mRNAs, eRNAs and PROMPTs are short and unstable due to high early termination rates and increased susceptibility to degradation by the RNA exosome[6,7], thus making their detection technically challenging.

Initiation of transcription in all the above regions depends on the efficient assembly of the pre-initiation complex (PIC) upstream of TSS and serine 5 phosphorylation (S5P) of Pol II C-terminal domain (CTD)[8]. After elongation of ~30–60 nucleotides of initiation-associated RNAs (start-RNAs)[7,9], Pol II is paused at PPP sites by negative elongation factors DSIF and NELF[2,10]. Signal-regulated phosphorylation of these factors and of Pol II CTD serine 2 (S2P) by P-TEFb is required for productive elongation[11–13]. It recently emerged that, if this step does not occur rapidly, start-RNAs are terminated[12,14], implying that Pol II turnover at PPP sites is high at steady state, and that replenishment of Pol II engaged in early transcription (initiation to PPP) is achieved through the continuous re-entry of pre-initiating Pol II into PICs[14,15].

The integrity of the genetic information encoded in DNA sequence is persistently challenged by a variety of genotoxic perturbations[16]. A plethora of DNA damage response (DDR) mechanisms have evolved to guarantee the detection and removal of different types of DNA lesions, limiting the probability of mutagenesis by adjusting to the cell's status and need for efficient recovery from DNA damage[16–18]. Nucleotide excision repair (NER) plays a vital role in sensing and removing a large panel of helix-distorting DNA adducts such as cyclobutane pyrimidine dimers (CPDs) induced by ultraviolet (UV) light, and benzo[a] pyrene guanine adducts induced by cigarette smoke[17]. Transcription-coupled NER (TC-NER) is promptly triggered by elongating Pol II molecules encountering DNA adducts and speeds up excision and repair in expressed loci[19–21]. In comparison, the second NER subpathway, Global Genome-NER (GG-NER), operates through the entire genome, but recognizes more stochastically helix distortions[17,21–23]. Importantly, given all the classes of transcripts defined above, it is estimated that the coverage of transcribed regions[24] potentially scanned by TC-NER expands to more than 50% of the genome, thus qualifying transcription as a major driving force in safeguarding genomic stability.

Although TC-NER depends on lesion-sensing potential by elongating Pol II molecules, transcription elongation has been shown to be transiently inhibited after UV irradiation[25–27] due to a proportion of Pol II molecules stalling at encountered DNA damages[25,28]. Moreover, depletion of the pre-initiating hypo-phosphorylated Pol II(hypo) isoform from chromatin shortly after UV irradiation[25,29,30] has led to the assumption that new transcription initiation events are transiently and globally repressed[21,29–33]. On the other hand, recent reports[25,26,34] have revealed a functionally essential stress-dependent increase in 5′ nascent RNA (nRNA) activity globally that depends on the UV-induced increase in active P-TEFb levels[35,36], and on the rapid dissociation of the NELF complex[37]. The ensuing fast and global release of de novo Pol II elongation waves from PPP sites into gene bodies boosts lesion-sensing activity and accelerates removal of DNA adducts by TC-NER in virtually all active mRNA genes[25]. Together, these findings substantiate the possibility that initiation of transcription might not be as severely affected by UV as previously believed.

Taking also into consideration recent evidence that supports the model of disengagement of a given Pol II molecule from DNA template after damage recognition[21,31,38], it is tempting to assume that ensuring continuity in transcription initiation may bring advantages in the repair process. We thus hypothesized that the apparent loss of pre-initiating RNAPII may not be due to the absence of RNAPII recruitment at TSSs, but rather due to a decrease in the dwell time of Pol II-hypo isoform at TSS, as justified by the concomitant increase in Pol II-ser5P and Pol II-ser2P downstream of TSS[25]. In this way, cells would be able to uninterruptedly feed the global release of scanning enzymes into transcribable sequences, and guarantee the detection of more lesions along genes' template strand (TS). Herein, we decipher chromatin dynamics genome-wide upon UV damage, and find a significant gain in accessibility (ATAC-seq) at the TSSs of virtually all active regulatory regions controlling mRNAs, PROMPTs, and eRNA expression. This phenomenon is underlined by the maintenance of active histone marks (H3K27ac), the lack of deposition of transcriptional silencing modifications (H3K27me3) at transcribed loci, and is correlated with the influx of Pol II into productive elongation. The paradoxical decrease in pre-initiating Pol II-hypo at these TSSs upon UV is elucidated by revealing that the presence of Pol II-hypo is rescued when PPP release is drug-inhibited. Accordingly, preserved production of start-RNAs after UV stress lies under the increased production of nRNA, and is prevented only after inhibition of transcription initiation. The identified genome-wide dependence of initiation rate on promoter-proximal pause-release dynamics explains the seamless recruitment/initiation of Pol II upon UV, in turn enabling efficient repair of the totality of the sequences encoding active regulatory regions and mRNAs.

## Results

**Chromatin accessibility increases at active regulatory regions upon UV.** To characterize the impact that UV irradiation might have on the chromatin landscape of transcriptional regulatory regions during the early recovery times (from 0.5 to 4 h), and how this could be linked to the widespread PPP release of elongating Pol II and the local increase in nRNA production downstream of TSS[25,26,34], we first determined the genome-wide changes in chromatin accessibility. The omni-ATAC-seq protocol[39] was implemented in our system involving UV-C irradiation (15 J/m²) of human skin fibroblasts synchronized in G1 (see "Methods" and also ref. [25]). We reproducibly measured chromatin accessibility before (NO UV) and 2 h after (+UV) irradiation (Supplementary Fig. 1a, b), performed peak calling and mapped a total of 106,052 accessible regions (ARs) across conditions (Supplementary Fig. 1b), and combined the replicates (see "Methods" for details). ARs were enriched at promoters and intragenic or intergenic regions with transcriptional regulatory function (TSSs, TSS flanks, and enhancers according to ChromHMM annotation, Fig. 1a–c; Supplementary Fig. 1c–e, "Methods"). Interestingly, we reveal a widespread increase (up to 1.71 average fold change (FC)) in chromatin accessibility after stress at 97.9% of promoter-, 94.6% of intragenic-, and 94.4% of intergenic ARs (Fig. 1b–d; Supplementary Fig. 1e, f).

We then selected differentially accessible regions (DARs) by applying stringent thresholds both in terms of FC ($\text{Log}_2$ FC > 1) and $P$ value ($P < 0.001$), and found that 6410 loci showed particularly increased chromatin accessibility upon UV (DAR-gain) (Fig. 1e, top panel). DAR-gain found at promoter regions

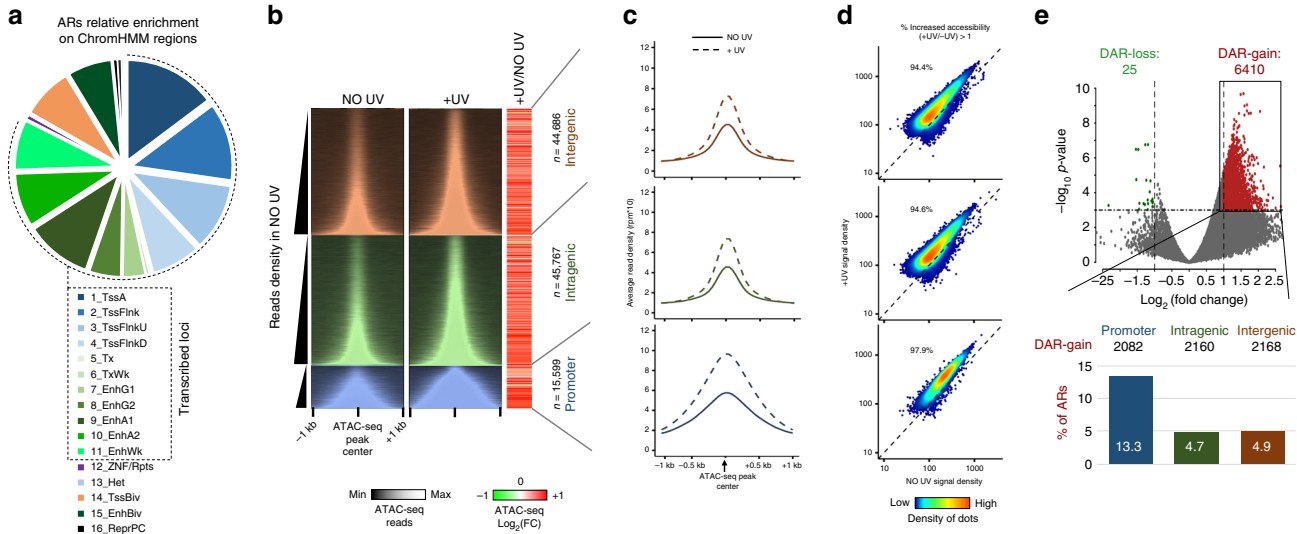

**Fig. 1 Increase in chromatin accessibility in response to mild doses of UV irradiation. a** Classification of ARs according to ChromHMM annotation. The dashed line represents active regulatory loci. **b** (Left panel) Heatmap of ATAC-seq reads in genomic regions 1 kb around ATAC-seq peak centers before (NO UV) and after UV (+UV, +2 h; 15 J/m$^2$), categorized according to their genomic position relative to RefSeq genes (intergenic, intragenic, and promoter peaks) and sorted by increasing read density (as determined before UV). (Right panel) Heatmap showing the log$_2$ fold change (log$_2$ FC) between +UV and NO UV read densities calculated in genomic regions 1 kb around ATAC-seq peak centers. **c** Average profile plots of ATAC-seq read densities of non-irradiated (solid line) and irradiated (dashed line) cells in intergenic (red), intragenic (green), and promoter (blue) regions. **d** Heat-density scatter plots comparing ATAC-seq read density before and after UV at all accessible regions (ARs) in intergenic, intragenic, and promoter regions, respectively. **e** (Upper panel) Volcano plot representing differentially accessible regions (DARs) between irradiated and non-irradiated cells. Regions with significantly increased (DAR-gain) or decreased (DAR-loss) accessibility are depicted in red and green, respectively. (Bottom panel) Proportion of DAR-gain loci in intergenic, intragenic, and promoter ARs.

represented 13.3% of all promoter ARs (Fig. 1e, lower panel), thus pinpointing towards a potentially functionally relevant chromatin opening at TSS regions. DAR-gain located at intragenic and intergenic loci (Fig. 1e) were linked to genes if they overlapped functional enhancers defined in FANTOM5 (see "Methods"). We found that genes associated with DAR-gain loci (either identified on their promoter or enhancers) were representative (adjusted (adj.) $P < 0.05$) of a number of biological pathways previously associated with DDR processes, including cellular response to stress, DNA repair, transcription regulation by TP53, and cell cycle checkpoints (Supplementary Fig. 2). In addition, we identified a broad range of many other significant Gene Ontology (GO) categories (163 in total, Supplementary Fig. 2), a result in line with the previously reported global PPP release of elongating Pol II waves at all active gene bodies upon UV irradiation[25].

**Chromatin marks linked to transcription remain stable after UV.** A number of studies have demonstrated that the turnover, modification, and/or degradation of histones around damage sites represent essential steps in conserved pathways that help cells deal with genotoxic stress[40,41]. However, especially in the case of UV-C-induced DNA damage, little is known about the post-translational modifications (PTMs) of histones around transcriptional regulatory regions. To better interpret the increase in chromatin accessibility and clarify its possible impact on genome-wide transcription dynamics, we studied the differential presence of two histone PTMs representative for the transcription status of associated chromatin: the silencing mark H3K27me3 and the activation mark H3K27ac[42,43].

We conducted ChIP-seq experiments with antibodies specific for both H3K27ac and H3K27me3 in NO UV and +UV (+2 h) conditions, and focused our analysis on TSSs of mRNAs and on a robust set of eTSSs, which are known to be functional and potentially transcribed in the investigated cell type according to

the FANTOM5 database (see "Methods"). We used the ChIP-seq data generated in this study (H3K27ac and H3K27me3), as well as previously published ChIP-seq data (Pol II-ser2P[25]) from the steady state (NO UV) condition, to determine subsets of active (presence of H3K27ac and Pol II-ser2P peaks over TSS), repressed (presence of H3K27me3 peaks over TSS), and inactive loci (no peak detected over TSS for H3K27ac, H3K27me3, and Pol II) (Fig. 2a, see "Methods") in our cell system. We associated the changes in histone marks and Pol II observed in these regions upon UV with ATAC-seq results. The increase in chromatin accessibility was detected at all active TSSs, which correspond largely to the promoters identified above (compare Figs. 1a and 2a, and see "Methods"), as well as FANTOM5-annotated active eTSSs upon UV (Fig. 2b, ATAC, 95% confidence interval (CI) excludes 0). This opening was in sharp contrast to the UV-induced global loss of Pol II-hypo at TSSs and eTSSs (Fig. 2b, Pol II-hypo, 95% CI excludes 0) observed 1.5 h after UV irradiation (8 J/m$^2$) (Fig. 2a, b; Supplementary Fig. 3).

Strikingly, we also found preservation (slight but not significant increase) of H3K27ac levels (Fig. 2a, b, 95% CI includes 0; Supplementary Fig. 3), and we observed no exchange of H3K27ac for H3K27me3 in response to UV at these active TSSs and eTSSs. Reciprocally, there was no loss of H3K27me3 for H3K27ac, and no gain of Pol II at repressed loci (Fig. 2a; Supplementary Fig. 3). Accordingly, the results of our genome-wide analysis were consistent with biochemical evidence obtained by histone acetic extraction followed by western blot analysis, showing that the global levels of H3K27me3 or H3K27ac remain fairly stable during the early period of recovery from UV stress (15 J/m$^2$) (Supplementary Fig. 3c, d).

We therefore conclude that depletion of detectable Pol II-hypo at TSSs and eTSSs does not occur due to repression of these loci by trimethylation of H3K27[43], or because of the loss of the activating histone mark H3K27ac[42].

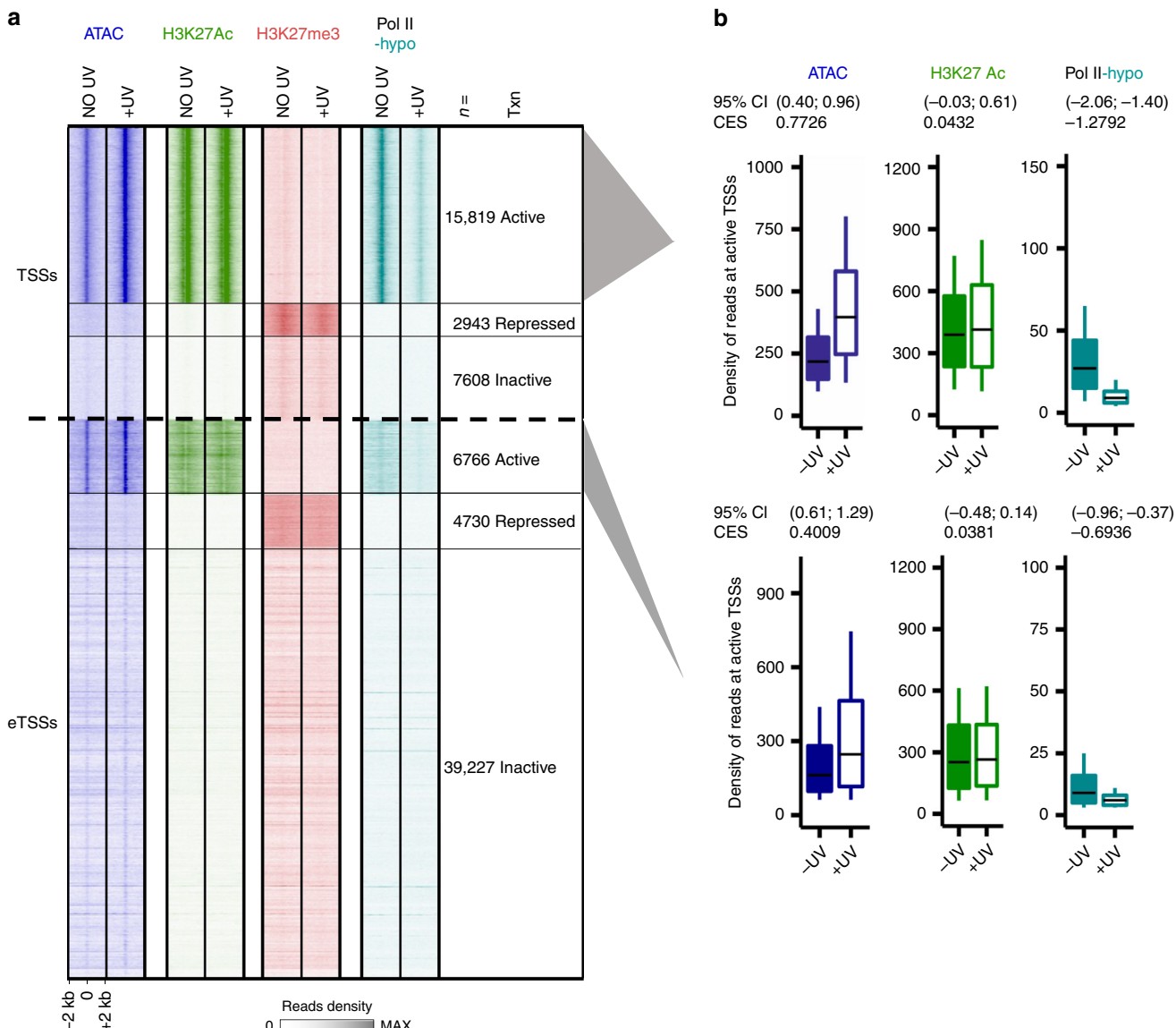

**Fig. 2 Histone modifications remain virtually stable upon UV damage. a** Heatmap depicting read densities for ATAC-seq, H3K27ac, H3K27me3, and Pol II-hypo ChIP-seq before (NO UV) and at 2 h post UV (+UV: ATAC and H3K27ac: 15 J/m², H3K27me3: 20 J/m²), for genomic regions 2 kb around active, inactive, and repressed TSSs and eTSSs, respectively. Data for Pol II-hypo are obtained from ref. [25] (at 1.5 h post UV with 8 J/m², see "Methods" for details). **b** Boxplots summarizing quantifications of ChIP-seq reads shown for active TSSs and eTSSs, respectively. Boxplots show the 25th–75th percentiles, and error bars depict data range to the larger/smaller value no more than 1.5 * IQR (interquartile range, or distance between the first and third quartiles). In all, 95% confidence intervals (CI) of mean differences between +UV and NO UV of log₂ counts were calculated for 10,000 samplings of 100 data points with replacement from each population. Effect sizes of log₂ counts between irradiated and nonirradiated samples were calculated by using Cohen's method (CES).

**Chromatin opening parallels Pol II transition into elongation.** To elucidate the functional advantage associated with increased chromatin accessibility in response to UV, we performed a thorough integrative analysis of our data in relation to previously published data sets (Pol II-ser2P from ref. [25] and Cap Analysis Gene Expression (CAGE)-seq from ref. [4], see "Methods"). First, we customized a genome annotation, which unambiguously pinpoints to the TSSs of mRNAs, PROMPTs, and eRNAs that do not overlap with regions possibly being transcribed through from neighboring/overlapping genes, promoters, or enhancers (see "Methods"). We then established three categories (Fig. 3a–c), as per previously suggested models[44]: first, active bidirectional promoter regions, which include the TSSs of mRNA–mRNA pairs transcribed in opposite directions (Fig. 3a); second, active unidirectional promoters, which include the TSS of a given

mRNA gene (+ or −) for which we could associate an expressed PROMPT in the antisense direction (Fig. 3b); third, active intergenic—as opposed to intragenic—enhancers to avoid potential contamination by interfering reads that derive from overlapping transcription of other active elements (Fig. 3c). Importantly, PROMPT and enhancer transcriptional activity were defined from available CAGE data for the skin and dermal fibroblasts (FANTOM5 consortium, see "Methods") that accurately determine transcript starting position (5′ end), abundance, and directionality of Pol II transcription in our model (Fig. 3a–c, CAGE). TSS loci were sorted by inter-TSS distance, which we defined as the distance separating TSSs and/or the summits of CAGE signals detected on the (+) and (−) strands (Fig. 3a, b; "Methods"). This allowed us to identify regions with overlapping (convergent, CONV) or non-overlapping (divergent, DIV)

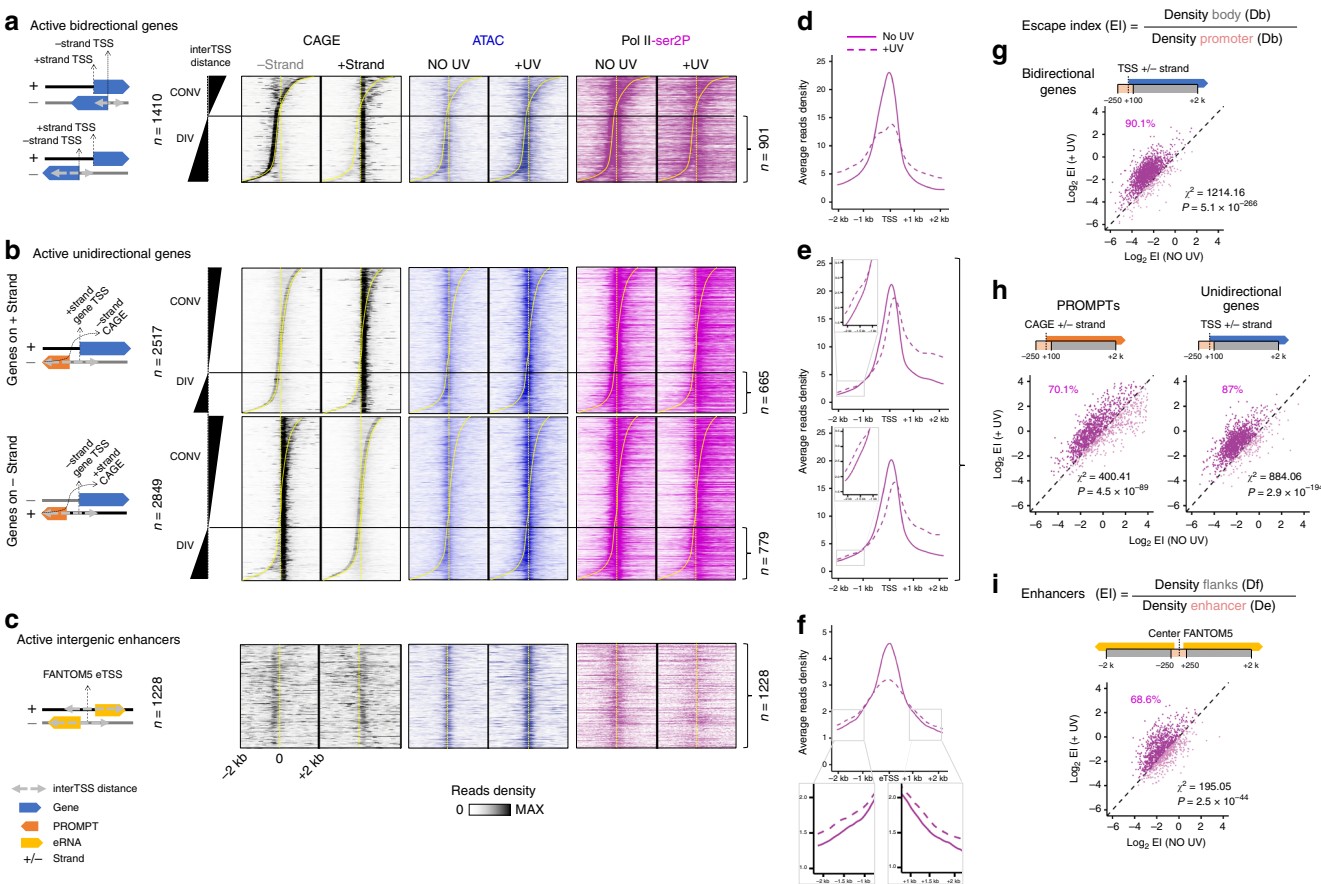

**Fig. 3 Release of Pol II from pausing sites in coding and non-coding transcribed regions upon UV stress. a** (Left) Scheme of convergent ("CONV", overlapping) and divergent ("DIV", non-overlapping) active bidirectional promoters expressing two mRNAs (blue arrows). (Right) Heatmap showing the distribution of CAGE (black; − and + strand separately) in the steady state, and ATAC-seq (blue) and Pol II-ser2P (purple) read densities, before (NO UV) and 2 h after UV (+UV) around +strand TSS (±2 kb). Loci are sorted by inter-TSS distance ((+ strand TSS)-(-strand TSS)). Data for Pol II-ser2P are obtained from ref. [25] and CAGE data from ref. [4]. Divergent loci correspond to inter-TSS >100 bp. **b** Same as in **a** but for active unidirectional genes TSSs, where PROMPTs (orange arrow) are transcribed in the opposite direction to the mRNA gene, from either the − strand (upper panel) or the + strand (bottom panel). Straight dashed lines indicate the position of mRNA TSS, and the sigmoidal dashed line indicates the variable relative position of CAGE PROMPT. Loci are sorted by inter-TSS distance (mRNA TSS - CAGE PROMPT) from the most convergent (mRNA and PROMPT overlapping, inter-TSS < 100 bp) to the most divergent (mRNA and PROMPT non-overlapping, inter-TSS > 100 bp) loci. **c** Same as in **a**, but for active intergenic enhancers expressing eRNAs (yellow arrow) in the opposite direction. **d**, **e** Average plots of Pol II-Ser2P before (solid line) and after UV (dashed line) on divergent categories defined in **a**, **b**. Insets represent a zoomed view. **f** Same as in **d**, **e**, but for all active intergenic enhancers. **g**–**i** Comparison of escape index (EI), for indicated categories of **d**–**f**. Percentages of loci with increased EI after UV (dark purple) are shown. Chi-square test between active bidirectional and inactive unidirectional genes (**g**), active and inactive unidirectional genes, and active and inactive PROMPTs (**h**), and between active and inactive enhancers (**i**) was performed to determine whether the observed number of genes with ΔEI >1 (number of genes with EI after irradiation greater than EI before irradiation) differs from the expected value purely by chance (see "Methods" for details).

transcription (Fig. 3a, b). By focusing on the latter category, we could study the dynamics of transcription at play only in each direction, without having to deal with potential interferences from overlapping regions.

Using this setup, we discovered that the UV-dependent increase in chromatin accessibility (Fig. 3a–c, ATAC) was paralleled to the transition of Pol II into active elongation (Fig. 3a–c, Pol II-ser2P), not only at flanking mRNAs (Fig. 3d, e) but also at adjacent PROMPTs and eRNA sequences (Fig. 3e, f), as shown by the loss in Pol II reads at TSSs and the gain of reads in downstream regions. These results were confirmed quantitatively by showing that escape index (EI) of elongating Pol II increased in the +UV condition in comparison with NO UV for 90.1% of bidirectional promoters (Fig. 3g, Chi-square test $P = 5.1 \times 10^{-266}$), as well as for 70.1% of PROMPTs (Fig. 3h, Chi-square test $P = 4.5 \times 10^{-89}$) and 68.6% of eRNAs (Fig. 3i, Chi-square test $P = 2.5 \times 10^{-44}$). We conclude that the PPP

release of Pol II upon genotoxic stress is synchronously triggered at all active transcription units, and coincides with increased chromatin breathing. These data extend the previously characterized transcription-driven genome surveillance mechanism[25] to essentially all active gene-regulatory regions, and give mechanistic insights into the synergy between the increase in chromatin accessibility and the transcriptional response observed upon UV.

**DRB rescues post-UV detection of Pol II in PIC.** We noted that although 63.65% of the transcribed genome shows reduction in transcription activity (coverage of the transcriptome with Log₂ FC (+UV/NO UV)) < 0, see "Methods") in line with earlier published data[45], a local increase in nRNA synthesis downstream of TSS of all active genes is detected during the UV-recovery phase[25–27,34]. This observation combined with the above findings on the UV-induced chromatin opening around virtually all active

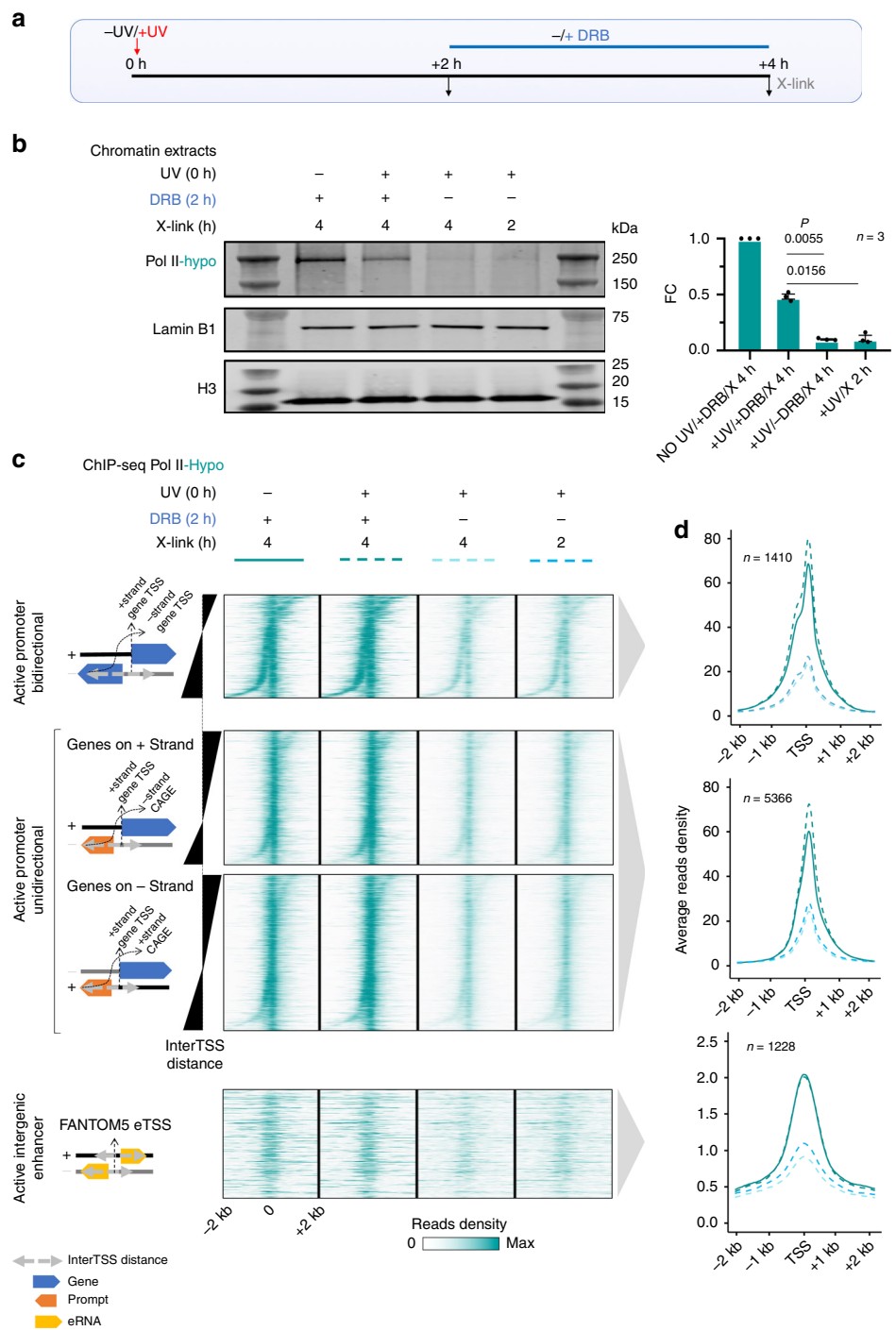

**Fig. 4 DRB rescues pre-initiating Pol II levels following DNA damage. a** Experimental timeline showing times of UV irradiation (15 J/m$^2$) and DRB treatment (see "Methods"). **b** Western blot analysis of chromatin extracts for Pol II-hypo levels as examined after employing the experimental strategy described in **a**. Lamin B1 and histone H3 were used as loading controls. Bar graph represents quantification of Pol II-hypo levels as compared with the NO UV/+DRB condition. Data shown reflect three independent experiments. Error bars represent S.E.M., and P values are calculated using two-sided Student's t test. **c** Heatmap of Pol II-hypo ChIP-seq read densities in genomic regions 2 kb around TSS for categories defined in Fig. 3 **a–c** after performing the combination of UV/DRB treatments described in **a**. **d** Average profile plots of read densities analyzed in **c**.

TSSs, PROMPTs, and eTSSs are hardly compatible with the previously suggested model of UV-induced global inhibition of transcription initiation. We thus searched for alternative reasons that could explain reduction of Pol II-hypo levels at active TSSs/eTSSs despite increased accessibility after UV.

We performed a set of experiments aiming to determine whether Pol II was actually recruited to TSSs upon UV (Pol II-hypo isoform

used as a proxy). First, as depicted in Fig. 4a, we irradiated cells with a UV (15 J/m$^2$) and left them to recover for 2 h, when the levels of Pol II-hypo are known to be severely depleted[25,29,30] (Supplementary Fig. 4a, b (DMSO NO UV vs DMSO +UV +2 h)). We then applied, or not, an inhibitor of Pol II release into elongation from PPP sites (DRB, see "Methods"). Cells were crosslinked (X) 2 h after the addition of DRB (or DMSO for the control cells). In accordance

with the above mentioned previous reports, in cells that were crosslinked 2 h after UV irradiation in the absence of DRB (+UV/X 2 h), or in cells that were crosslinked 4 h after UV irradiation, and had been incubated with DMSO for the last 2 h (+UV/−DRB/ X 4 h), we detected only minimal levels of pre-initiating Pol II in total chromatin extracts or at TSSs, PROMPTs, and eTSSs, as revealed by Western blot analysis (Fig. 4b) and ChIP-seq (Fig. 4c, d), respectively. In contrast, when cells had been incubated with DRB for the last 2 h before being crosslinked at 4 h after UV irradiation (+UV/+DRB/X 4 h), we observed a significant rescue of pre-initiating Pol II (hypo) levels in total chromatin (Fig. 4b, two-sided Student's $t$ test $P = 0.0055$ compared with "+UV/-DRB /X 4 h" and $P = 0.0156$ compared with "+UV/X 2 h"). The restoration of pre-initiating Pol II levels was even more pronounced when we focused on the occupancy on active TSSs, PROMPTs, and eTSSs, where average read densities detected by Pol II-hypo ChIP-seq after DRB treatment (+UV/+DRB/X 4 h) matched the control NO UV levels (NO UV/+DRB/X 4 h) (Fig. 4c, d). Therefore, even by blocking the stress-triggered transition of Pol II molecules from PPP sites into elongation at 2 h post UV, when the prior-to-UV Pol II-hypo levels were almost completely depleted, we were able to reveal the underlying continuous de novo recruitment of Pol II-hypo molecules in PICs.

We also applied DRB just before and for 2 h after UV (15 J/m$^2$, Supplementary Fig. 4a), and found a limited loss of pre-initiating Pol II in chromatin extracts upon UV (Supplementary Fig. 4b, c, two-sided Student's $t$ test $P = 0.0145$). This result was corroborated by ChIP-qPCR experiments (performed on the same chromatin extracts used above), as DRB prevented the UV-induced reduction in occupancy of Pol II-hypo at promoter/TSS-proximal regions of six active genes (Supplementary Fig. 4d, two-sided Student's $t$ test $P = 0.003$ for DMSO, while $P = 0.2876$ (non-significant) for DRB).

We thus conclude that the genome-wide UV-induced PPP release of Pol II molecules into elongation accelerates the transition into initiation of the next-to-be-recruited Pol II-hypo molecules, limiting the dwell time of this isoform at essentially all active TSSs, PROMPTs, and eTSSs.

**Increased RNA synthesis from active TSSs upon UV.** As UV irradiation does not inhibit neither the recruitment of Pol II-hypo into PICs nor Pol II escape into elongation, we next verified that newly synthesized RNA molecules were detected at the beginning of transcribed regions. We took advantage of our and other nRNA-seq data[25,27], and we examined if the previously characterized global increase in EU- or Bru-labeled RNA reads at the beginning of genes (see Supplementary Fig. 4 in ref. [25]) could originate from increased Pol II initiation at active TSSs (Fig. 5a, b; Supplementary Fig. 5a, b), as suggested before[27]. In particular, at unidirectional promoters, we confirmed that nRNA synthesis was increased in the mRNA direction, but we also found a concomitant increase of nRNA production in the antisense, PROMPT direction. Similarly, we found widespread gains in eRNAs synthesis, which emanate equally in both directions from active eTSSs (Fig. 5a, b; Supplementary Fig. 5a, b). Identifying labeled nRNA even at short transcripts such as PROMPTs and eRNAs confirms active labeling close to TSSs, and validates the fact that regions directly downstream of TSSs get de novo transcribed during the post-UV period. Taken together, these data demonstrate that the continuous recruitment of Pol II-hypo molecules (see Fig. 4) and their fast transition into initiation/productive elongation (see Fig. 3), during the recovery from genotoxic stress, is accompanied by de novo synthesis of RNA directly downstream of TSSs.

To further verify initiation activity during UV recovery, we exploited the possibility to track start-RNAs, which directly inform on the amount of dynamically engaged Pol II located within the initially transcribed sequence (approximately the first 100 nucleotides[7]). We followed the experimental procedure depicted in Fig. 5c, and applied, or not, transcription elongation (DRB) or initiation (triptolide, TRP) inhibitors 2 h post UV (15 J/m$^2$). For each condition, we isolated small RNAs by size selection (<200 nucleotides), and we ligated an RNA–DNA linker to their 3′ ends. Reverse transcription (RT) was performed using a universal primer annealing to the linker sequence as previously described[7]. Subsequently, locus-specific qPCR reactions were performed in order to compare, in a quantitative way, the levels of start-RNAs at representative active loci for which we had identified Pol II-ser2P ChIP-seq or nRNA-seq signal (see "Methods"). Our results revealed that start-RNAs could be detected after UV treatment, validating the fact that initiation still occurs during the UV-recovery phase (Fig. 5d, +UV/−DRB). Similar results were obtained in the presence of the transcription elongation inhibitor (Fig. 5d, +UV/+DRB). However, the opposite was found after inhibiting transcription initiation by TRP, which as expected led to a clear reduction of start-RNAs (Fig. 5d, +UV/+TRP, two-sided Student's $t$ test, $P = 0.0037$ compared with "NO UV/+DRB", $P = 0.0016$ compared with "+UV/−DRB", and $P = 0.0009$ compared with "+UV/+DRB"). Together, these results consolidate further the evidence of the non-stop recruitment and initiation of Pol II at TSSs after UV irradiation.

**Equal levels of Pol II-hypo at PICs prime for uniform TC-NER.** Next, we took advantage of XR-seq data (eXcision-Repair sequencing)[20], which precisely and exclusively pinpoint the location and levels of transcription-dependent repair (TC-NER pathway) when the assay is performed in GG-NER-deficient cells (xeroderma pigmentosum (XP)-C cells). Given the strand specificity of the assay, we considered only the excision of CPD damages from template (non-coding) strand (TS) for mRNAs, PROMPTs, and eRNAs, which corresponded to the + (blue) or the − (red) strand of the genome (Fig. 6a, b), depending on the transcript orientation. Upon correlation with CAGE, we found that onset of TC-NER coincided with CAGE reads location, confirming the fact that TC-NER (triggered by damage-arrested Pol II molecules[20]) and CAGE[4] accurately locate active TSSs (Fig. 6b (compare with Fig. 3a–c, left), Fig. 6c, d). As expected, repair efficiency was equal in each direction for bidirectional active promoters (Fig. 6b–e). This result was also in line with Pol II-hypo ChIP-seq data showing equivalent amounts of Pol II recruitment at PICs in both directions (Supplementary Fig. 6a, b), and CAGE data indicating strand-balanced production of capped mRNAs (Fig. 6e, CAGE, boxes centered around Log$_2$ FC $= 0$). Nevertheless, we note that the variability between both directions was strikingly less for TC-NER and Pol II-hypo than for CAGE (Fig. 6e (proportion of non-significant F-Tests: $P = 0$), Supplementary Fig. 6b (top panel, proportion of non-significant F-Tests: $P = 0$)).

Next, we further investigated repair of PROMPTs and enhancers, a phenomenon previously observed, but hardly explained[20,46]. We quantified strand-specific repair upstream and downstream of unidirectional promoters, and found that repair activity at unambiguously resolved divergent PROMPTs was stronger than expected from CAGE levels (Fig. 6b–d, DIV). Indeed, XR-seq read density was not correlated to the steady state levels of CAGE at those loci (Pearson correlation coefficient (PCC) $= 0.1343$). Also, FCs of TC-NER reads between mRNAs and PROMPTs were much smaller than those for CAGE (Fig. 6f, 95% CI excludes 0), thus matching the UV-independent Pol II-hypo uniformity (Supplementary Fig. 6a, b). Similarly, TC-NER levels on TS of eRNAs were higher than anticipated. Indeed, the densities of eRNA

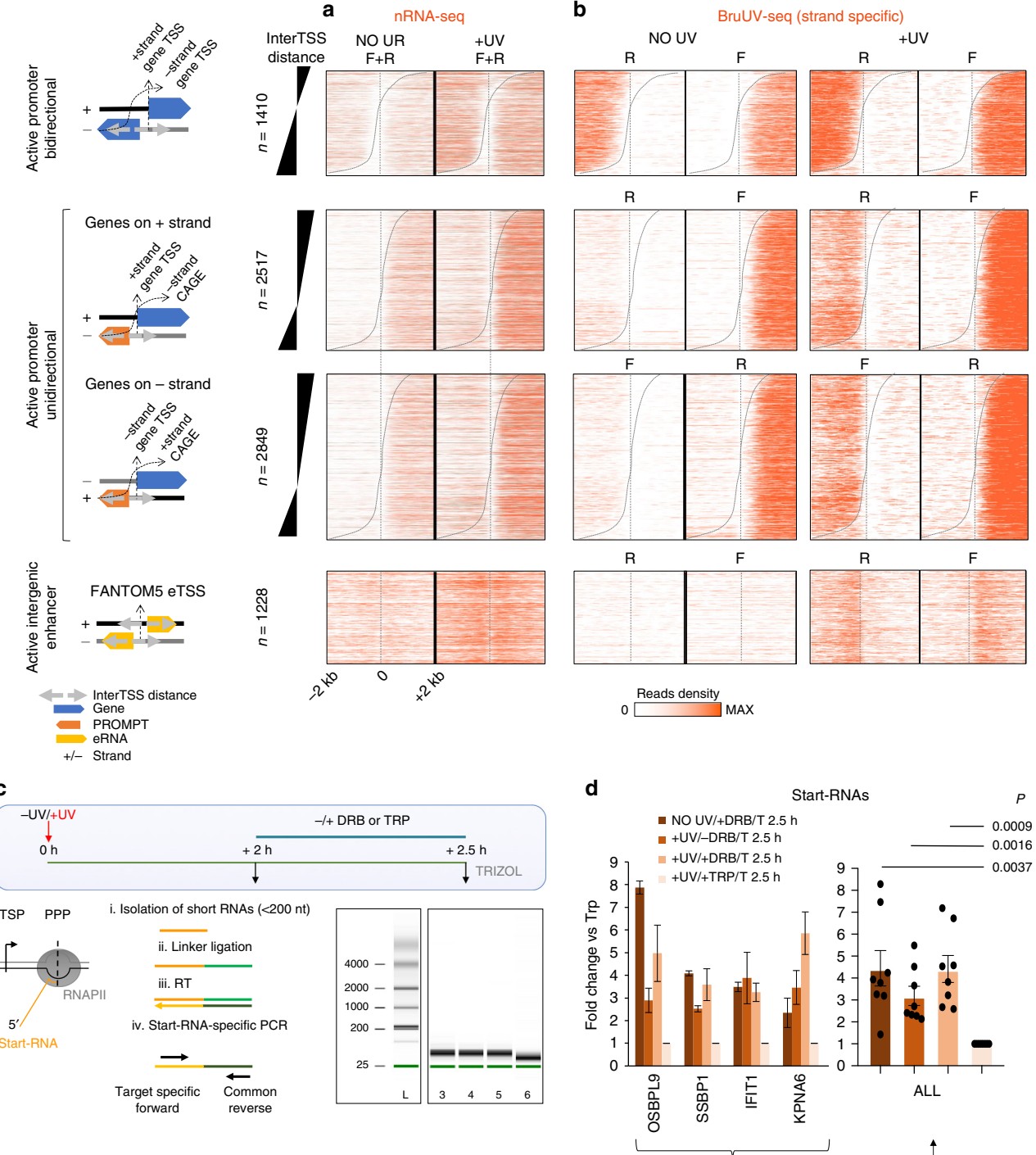

**Fig. 5 De novo and increased RNA synthesis from all TSSs upon UV exposure. a**. Heatmap of nascent RNA (nRNA) read densities (NO UV and +UV (60 min, 20 J/m²) data obtained from ref. 25 before and after UV, in genomic regions 2 kb around TSS, for categories defined in Fig. 3a–c (see "Methods" and Supplementary Fig. 5 for timeline). F: forward (+) strand, R: reverse (−) strand. **b** Same as in **a** for strand-specific BruUV-seq (NO UV and +UV (30 min, 20 J/m²) data obtained from ref. 27). **c** (Upper panel) Experimental outline. Cells were treated (or not) with UV, and were left to recover normally for 2 h. In turn, DRB, TRP, or DMSO was added, and after 30 min, cells were disrupted by Trizol addition. (Lower panel, left) Methodology followed for the detection and quantification of gene-specific start-RNAs (for details see "Methods"). (Lower right) Agilent RNA 6000 Nano Bioanalyzer traces showing size distribution of RNA samples after preparation of a separate small-sized RNA fraction. L: RNA ladder (size in nucleotides), 3–6: small-sized RNA fraction of samples analyzed in **d**. **d** qPCR analysis of start-RNAs. Bar chart illustrating FC (compared with TRP treatment), for each gene tested (left) and for the average FC of all genes (right). Error bars represent S.E.M., and P values are calculated using two-sided Student's t test.

XR-seq reads were similar to those of mRNAs (Fig. 6b–d), and contrasted with the very low CAGE signal detected at these loci (Fig. 3a–c, left). Therefore, balanced Pol II-hypo loading in PICs at all classes of transcripts, during steady state or upon stress, allows

for equal initiation events and mirrors the homogeneous levels of XR-seq detected in these regions. Taken together, our results demonstrate that the widespread continual initiation and release into productive elongation of Pol II waves maximize repair activity,

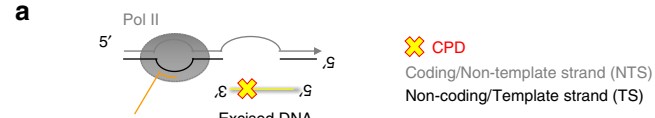

regardless of prior-to-UV transcript expression level at all kinds of active regulatory regions (mRNAs, PROMPTs, and enhancers).

**Continuous initiation drives TC-NER to completion.** We next assessed the biological relevance of continuous transcription initiation from active regulatory regions during the UV-recovery period. We have reported previously[25] that in the absence of a UV-triggered PPP release of elongating Pol II waves, the scanning activity of pri-elongating (e.g., already elongating prior to UV) Pol II molecules is not sufficient to enable recognition of damaged thymidine dimers (TpT sequence, abbreviated as TT) in the totality of the transcribed genome. In accordance, a recent study reports that inhibition of transcription elongation with DRB

limits the extent of excision activity detected by CPD XR-seq[38]. Thus, sending Pol II molecules from TSS into gene bodies after UV is of pivotal importance to maximize the detection of TTs on active TS.

We quantified the effect of various regimes of DRB (see scheme in Fig. 7a) on the dynamics and distribution of TC-NER (CPD XR-seq data from XP-C cells[38]) around TTs and around TSSs of mRNA genes, PROMPTs, and enhancers (Fig. 7a–d, respectively, see "Methods" and ref. [25]). We observe that when DRB was applied immediately after UV, repair of TTs located from TSS to upstream of the +1 -h wave front (WF) of UV-released Pol II (as obtained in the non-DRB control) are seriously affected (Fig. 7a–c, DRB + UV + 1 h, clusters 0-II).

**Fig. 6 TC-NER is homogeneous at all transcribed regions. a** Scheme representing the orientation and nomenclature of the DNA strands for TC-NER-specific XR-seq analysis. **b** Heatmaps of re-analyzed TC-NER excision reads of GG-NER-deficient cells[20] (XP-C), which are detected on template strand (TS) + (blue) or − (red) strand of the genome, 2 kb around TSSs for categories defined in Fig. 3a–c ("Methods"). Dotted gray lines denote the CAGE summits detected on each strand; the dark-blue dotted lines indicate positions 500 bases downstream of CAGE summits for the corresponding strand. **c** UCSC genome browser snapshots of representative loci for categories defined in **b**. **d** Average profiles of read densities derived from **b**: only the divergent (DIV) loci were considered. **e** (Left panel) Scheme representing the range used for calculating Log₂ FC of reads between + strand and −strand at divergent loci. (Right panel) Boxplots showing quantifications of the ratio of reads between directions (indicated window sizes and borders) at bidirectional promoters for CAGE reads (shown in Fig. 3), and TC-NER-specific XR-seq reads shown in **b**. Boxplots show the 25th–75th percentiles, and error bars depict data range to the larger/smaller value no more than 1.5 * IQR (interquartile range, or distance between the first and third quartiles). Two-sample F-tests were conducted for each of 10,000 sampling pairs of 100 data points with replacement from each population to test for significant differences between sample variance. The calculated P expresses the percentage of the non-significant F-tests (F-test P >= 0.05) out of all tests. **f** Comparison of CAGE and TC-NER-specific XR-seq reads as in **e**, but between divergent non-overlapping mRNA and PROMPTs (indicated window sizes and borders). In all, 95% confidence intervals (CI) of mean differences between log₂ counts of tested conditions were calculated for 10,000 samplings of 100 data points with replacement from each population. Effect sizes of log₂ counts between data sets were calculated using Cohen's method (CES).

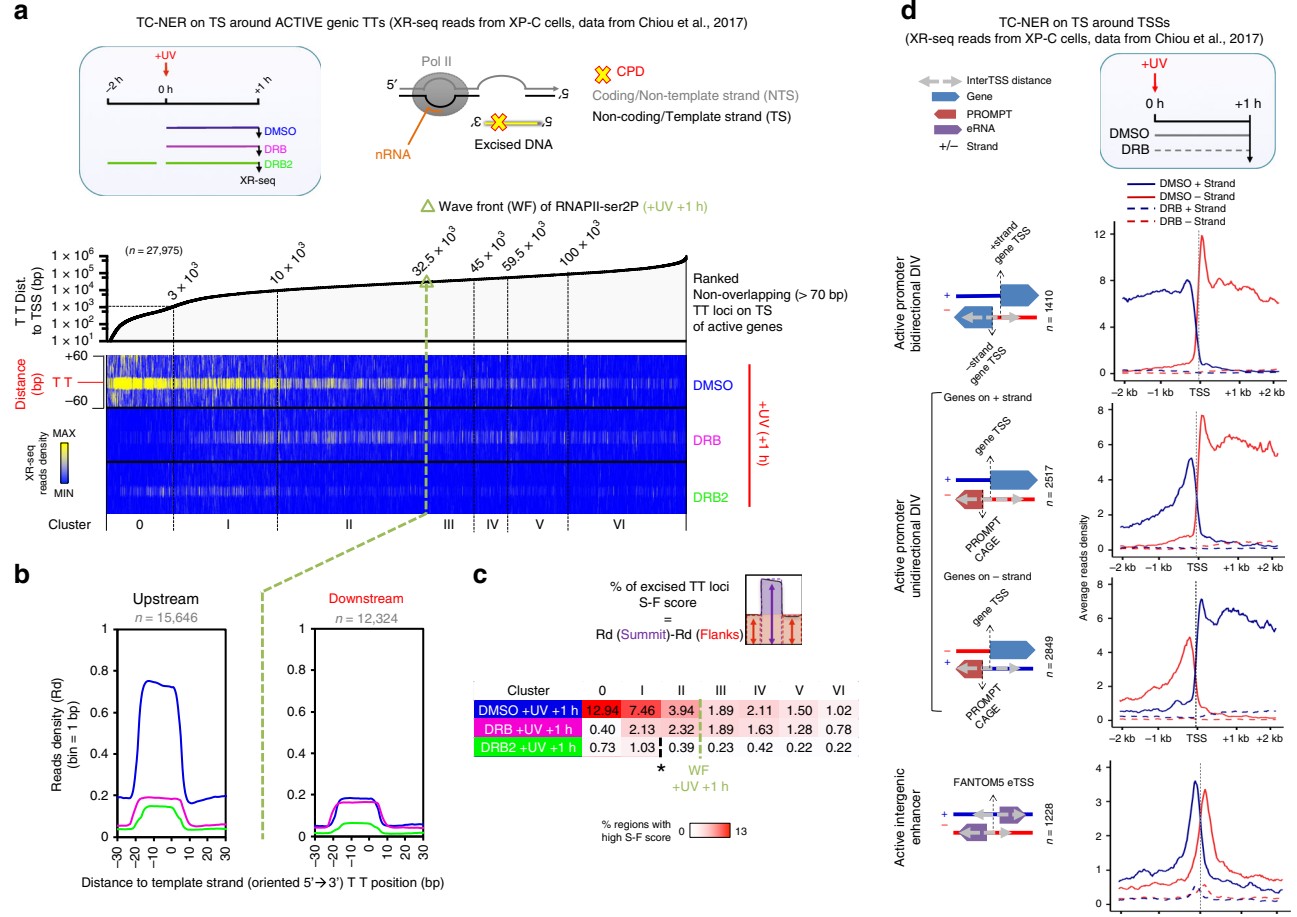

**Fig. 7 TC-NER activity monitored with various regimes of DRB reveals the need for continuous transcription initiation. a** Heatmap depicting the distribution of excised DNA fragments (XR-seq) derived from TC-NER activity (XP-C cells). Reads were aligned around TT loci of the transcribed strand, localized on active genes. Data were obtained from ref. [38]. The box in blue border (left) illustrates the experimental timeline followed and the respective drug treatments. **b** Average plots of read densities (Rd) showed in **a** but for clusters upstream (0, I, and II) and downstream (III, IV, V, and VI) of Pol II-ser2P wave front (WF) as defined in ref. [25] for +UV (+1 h) condition. **c** Plot showing the percentage (%) of excised TT loci, as calculated by the difference between Rd at summit (S) and Rd at flanks (F) (S–F score) from all clusters presented and analyzed in **a** and **b**. WF is indicated, and asterisk indicates where drastic decreases in the XR-seq signal occur in DRB2 experiments. **d** Average profile of read densities of XR-seq signal derived from XP-C cells. Dashed lines illustrate the XR-seq signal in + (blue) or − (red) strand, in the presence of DRB (see "Methods" for details).

Accordingly, allowing only a limited number of pri-elongating molecules to be launched just before the irradiation, while preventing de novo Pol II release after stress (Fig. 7a–c, DRB2+ UV + 1 h) highlights how an insufficient feed in Pol II impairs TC-NER activity at both distal and proximal transcribed regions (compare signal before and after asterisk positions in Fig. 7 a, c).

We reasoned that the high extent of ongoing repair activity detected without DRB (UV + DMSO) is the result of the continuity in Pol II initiation. To confirm this directly, we mapped XR-seq reads around TTs located on TS 1, 4, and 8 h after UV in WT cells (data obtained from ref. [46], see "Methods"). We found that significant levels of transcription-dependent

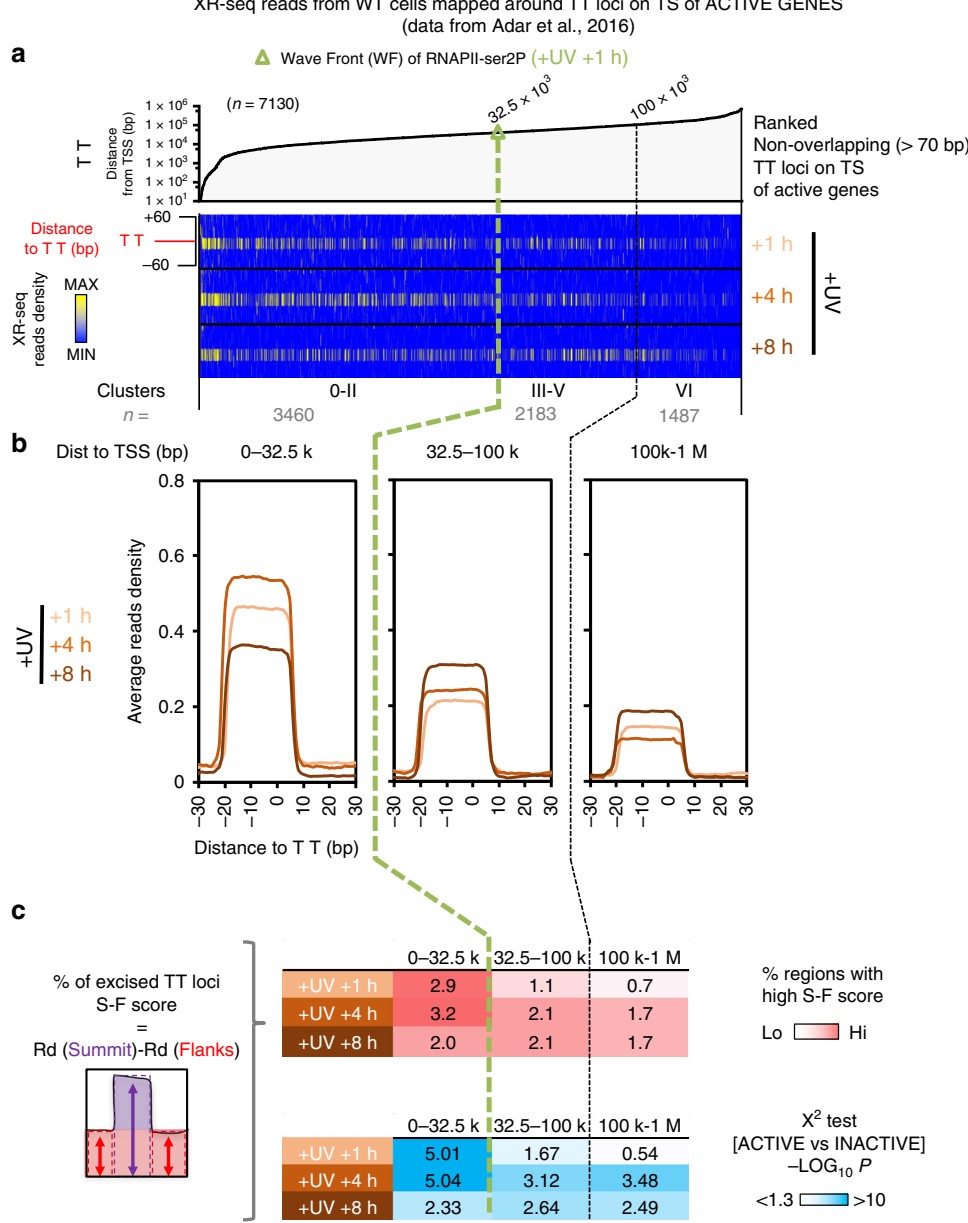

**Fig. 8 Continuous initiation maintains TC-NER efficiency during the whole recovery period and across the whole transcribed region. a** Distribution of XR-seq reads derived from WT cells at 1, 4, and 8 h post UV irradiation. Reads were aligned on TT loci of the transcribed strand (TS) of active genes. Data were obtained from ref. [46]. Pol II-ser2P wave front position (WF = 32.5 kb) was defined at +UV (+1 h) in ref. [25]. **b** Average plots of read densities shown in **a** for the respective clusters. **c** Heatmaps showing the percentage (%) of excised TT loci, as calculated by the difference between Rd at summit (S) and Rd at flanks (F) (S-F score) from all clusters presented and analyzed in **a** and **b** with high "S-F" scores (>threshold = average "S-F" + 3 × SD, as calculated for the control exon start regions, see ref. [25]) (see "Methods" for details).

excision activity were maintained at lesions located directly downstream of active genes TSSs at 4 and 8 h into the recovery process (compare Fig. 8a–c and Supplementary Fig. 7a–c). Notably, it also appeared that the bulk of the excision activity on TS of active genes shifts overtime (+8 h) from the proximal to the distal part of long genes (Fig. 8a–c, clusters III–VI; Supplementary Fig. 7d).

Overall, these results reveal that a large extent of the transcription-driven repair activity detected after UV is due to the ongoing entry of Pol II molecules at TSSs, which can scan farther and farther lesions along the gene bodies. Our analysis highlights the advantage of a continuously supplied transcription-dependent repair process (Fig. 8 and Supplementary Fig. 7d, TS of active genes) over slower CPD detection capabilities of GG-NER,

which was detected at significantly lower levels across all regions on NTS of active genes and on both strands of inactive genes (Supplementary Fig. 7d).

## Discussion

In this study, we provide quantitative insights into the molecular processes underlying the major transcription-coordinated cellular response that is activated in human cells upon genotoxic stress[25–27,34,35,37]. The establishment of precise maps of chromatin state helped us to query in detail the impact of transcription on DNA repair activities at important functional regions, including PROMPT and eRNA loci. Our results support a model of continuous transcription initiation that can feed the widespread

UV-triggered escape of Pol II into the elongation, enabling long-lasting efficient DNA lesion scanning of the whole transcribed genome.

The finding that an increase in chromatin accessibility parallels the conservation of H3K27ac-modified nucleosomes at the flanks of already opened regions in response to UV irradiation is compatible with reports showing that there can be a significant gain in DNA accessibility without changes in nucleosome occupancy during rapid transcriptional induction[47]. Notably, the maintenance of H3K27ac at these sites prevents the imposition of repressive trimethylation at active loci (see Fig. 2), in accordance with the rule that H3K27ac and H3K27me3 are mutually exclusive[48]. Moreover, finding more active transcription at these loci complies with prior reports, suggesting that an increase in gene expression is associated with surges in chromatin accessibility[49], and that the presence of nRNA inhibits the recruitment of H3K27me3-catalyzing polycomb-repressive complex 2 (PRC2) at active genes[50]. Our results obtained using a range of mild UV doses (8–20 J/m$^2$), and focusing on the early phase of recovery (1-4 h), support the fact that chromatin relaxation and maintenance of H3K27ac has an active role in the repair of transcription-blocking lesions, and give substance to previously observed rather low increase in acetylation of histones in similar conditions[51]. In contrast, drastic chromatin remodeling observed in murine cells at a later time during recovery (6 h) when much higher doses (80 J/m$^2$) were used[52] fits with findings signifying that cellular response to UV depends on the exposure dose[53,54]. In other words, when cells deal with unmanageable levels of damages, they need to implement more radical expression changes, which are required for the associated fate of programmed death[55,56], a protective mechanism limiting the risk of malignant transformation.

Our analysis takes advantage of a high-resolution strand-specific map of TSSs for coding and non-coding (enhancers and PROMPTs) loci, and supports the idea that bidirectional transcription of divergent RNAs arises from two distinct hubs of transcription initiation (PICs), located within a single nucleosome-depleted region (NDR)[8,57,58]. Indeed, for bidirectional mRNAs and mRNA-PROMPTs, the binding of Pol II-hypo occurs at both edges of highly accessible regions (see ATAC-seq vs Pol II-hypo in Supplementary Fig. 6a, c), which correspond to single NDRs flanked by H3K27ac nucleosomes (see arrows in Supplementary Fig. 6a, c). These observations also extend the evidence supporting the claim that enhancers and PROMPT PICs are organized in a similar manner to gene PICs[8,13]. Finally, the observed differences in transcript levels between PROMPTs and mRNAs (see Fig. 5) are probably not due to differences in Pol II-hypo recruitment (see Supplementary Fig. 6a, b, bottom), but rather due to differences in the frequency of premature termination at PPP sites and/or differences in degradation of PROMPT RNAs by the RNA exosome, which is known to be inhibited upon UV stress[59,60].

By uncoupling TSSs of mRNA genes from those of PROMPTs and enhancers, we reveal that P-TEFb-dependent release of elongating Pol II from PPP sites extends to all actively transcribed regions (see Fig. 3). Interestingly, a growing number of studies have reported data to suggest that (i) UV irradiation preferentially inhibits elongation, rather than transcription initiation[25–27,34], (ii) P-TEFb and NELF are important regulators of UV response[35,37,61], and (iii) although elongation gradually decelerates due to the encounter of Pol II with DNA lesions, significant initiation/early elongation activity is observed in the first thousand bases of actively transcribed regions[25,26,34], a characteristic that has also been used for the identification of active TSSs genome-wide after UV[27]. These features are consistent with our finding that new Pol II molecules are constantly recruited to PICs post-UV (see Fig. 4), suggesting that a pool of non-engaged polymerases is still available, and that they promptly

proceed into initiation of start-RNAs and subsequently into elongation of longer nRNAs (see Fig. 5). Considering that Pol II ChIP-seq density depends, among others, on the epitope residence time at a given genomic locus[62], and that Pol II molecules recruited in the PIC are readily phosphorylated upon UV[25,29,30], we propose that the rapid exchange of Pol II isoforms after UV irradiation represents a perfectly plausible cause for the decreased ability to detect Pol II-hypo molecules at TSSs upon UV. It also accounts for the increase in EI, and could explain the gain in accessibility around TSS (see Figs. 2–4, Supplementary Fig. 6, and ref. [25]). Such a model explains previously published data concerning the persistence of sufficiently important levels to be worthy of attention, of (i) PIC/basal transcription factors in nuclear extracts[30] or upstream of genes' TSSs (TFIIB)[33] and (ii) nRNAs at the beginning of genes[25–27,34] upon UV.

We note that excision fragments (from XR-seq) are distributed more homogeneously at sense (mRNA) and antisense (PROMPT) strands of unidirectional TSSs, and at enhancers, than it could be predicted from the CAGE levels (see Fig. 6). This finding reinforces the possibility that efficient repair at stable and unstable transcripts is primed by the uniform recruitment of Pol II-hypo at all classes of PICs in the steady state (see Supplementary Fig. 6). Remarkably, UV induces continuous and uniform transition into initiation (see Fig. 4), and constantly feeds a long-lasting PPP release of lesion-sensing transcribing Pol II into the transcriptome (see Figs. 3 and 5).

This concept was further validated by re-analyzing an experiment mapping excision repair of CPDs (XR-seq) after inhibition of PPP release[38], and thus a fortiori preventing the beneficial effects of Pol II initiation. Indeed, upon DRB treatment, we find a drastic impairment of CPD excisions at the beginning of all classes of active transcripts (see Fig. 7d), and in particular at TTs located within the distance normally covered by de novo released Pol II molecules, without inhibition, at 1 h post-UV (low XR-seq signal for DRB vs DMSO in cluster 0-II, see Fig. 7a, c). In addition, the decrease in the percentage of excision in cluster II (between 10 kb and WF) in DRB2 experiment (see Fig. 7 a–c, DRB2 vs DMSO) indicates that continuity in initiation post UV is crucial for the repair of sequences located toward the 3' end of longer genes. Taking into consideration that only one Pol II molecule can be accommodated per PPP site at each actively transcribed loci at the time of irradiation, applying DRB after UV mainly restricts the benefits related to the continuous recruitment, initiation, and release into elongation of multiple polymerases throughout the recovery period.

Critically, such a scheme is compatible with the evidence that continuous nRNA-seq signal can be detected uninterruptedly between 2 and 12 h post UV from the start to the end of genes[26]. We propose that keeping initiation active maximizes the probability to repair quickly all lesions located on the TS. Our model is compatible with the idea that even if a Pol II molecule is ubiquitylated and degraded after the recruitment of the NER complex[21,31,38], the amount of Pol II still normally recycling after termination from all the short genes (which contain less or no lesions) will provide sufficient trailing polymerases for scanning lesion-containing genes. In this way, this mechanism can sense efficiently one after the other the more distal lesions even in longer genes, and gives an alternative biological perspective to presently favored mode of action[21]. In agreement, we detect TC-NER activity on TS directly downstream of TSSs 4 and 8 h after UV. In addition, repair of distal damages in long genes is detected only when DRB is not applied after UV (compare clusters III–VI with DRB2 and DMSO in Fig. 7a–c), and intensifies at 8 h in normal conditions (see Fig. 8 clusters III-VI; Supplementary Fig. 7d, distal).

Increase in TC-NER at regulatory regions has also been observed in E. coli[63], and is compatible with the idea that the act of

antisense transcription over regulatory regions exerts a meaningful biological function[64] conserved through evolution. Indeed, these DNA sequences may serve as binding sites for transcription factors, or encode target sites for RNA-binding proteins, enabling accurate regulation of topologically associated mRNA genes[58,65]. Given the effect of DNA repair on the landscape of somatic mutations in cancer tissues[17,18], surveillance of these vital sequences impacts on cell's fitness. We propose that our model could account for the low levels of substitutions recently observed upstream of genes' TSSs and around DNAse-hypersensitive (DHS) sites[66,67]. As such, transcription of non-coding regulatory loci could serve to keep their transcription factor-binding sequences (TFBS) in check.

Recent advances in the field of transcription regulation point to the fact that activation of paused genes is mediated through switching from a premature termination state of Pol II at PPP sites to a processive elongation state[11,14,15], implying that continuous cycles of initiation followed by rapid premature termination are required for fast transcriptional induction[14]. Our results, showing that persistent initiation guarantees a prolonged transcription-coupled NER across the whole transcriptome, are functionally linked to the fact that DNA damage-triggered widespread PPP release of a given Pol II is sufficient to drive immediate initiation of the next Pol II (see Figs. 4 and 5; Supplementary Figs. 4 and 5). In other words, the clearance rate of Pol II from TSSs highly depends on PPP status. These findings are favorable to the emerging concept that Pol II promoter-proximal pausing has an inhibitory effect on initiation[68–70], and highlight how this mechanism can function across the whole transcriptome. At the same time, our results provide a compelling physiological relevance to why cells could gain from firing initiation continuously, as a balance between promoter-proximal termination and escape into elongation allows efficient dynamic responses to stimuli or genotoxic stress.

## Methods

**Cell culture and treatments**. Cells used in this study were VH10 hTERT-immortalized human skin fibroblasts, and were cultured, synchronized by low-serum starvation, and released in full medium as described previously[25], unless stated differently. When applied, 5,6-dichloro-1-β-D-ribofuranosylbenzimidazole (DRB, Calbiochem) and triptolide (called TRP, Invivogen) were used in a final concentration of 100 µM and 125 nM, respectively, and they were added directly in growth media at the indicated times. Cells were irradiated with UV-C (254 nm, TUV Lamp, Philips) (8–20 J/m² as indicated) and left to recover for indicated times (see figure legends and below).

**ChIP-seq**. ChIP-seq was performed as previously described[25] with minor changes (see Supplementary Methods for details). Cells were mock-treated (NO UV) or treated with UV (+UV) with 15 J/m² except if otherwise stated. Treated cells were left to recover for 2 h before harvesting or as indicated on the timeline (Fig. 4).

The antibodies used for ChIP were the following: H3K27ac (ab4729, Abcam), H3K27me3 (07-449, Millipore), and 8WG16 (Pol II-hypo) (05-952, Millipore). The primers used for ChIP-qPCR experiments were the following (5′ to 3′, F: forward, R: reverse, ChIA neg was the negative primer): SSBP1_F: GTGAGGGAGGAAGG GATAGC, SSBP1_R: AGGGCCAGACACCTACACAG, OSBPL9_F: ATTGGCGG CTCCCAAGAT, OSBPL9_R: GCATTGTAGTCCAGCACGAA, TRPM7_F: CCC AGGGAAACCTTCTCAG, TRPM7_R: TCGCACAATTATGAAAGACTCG, MY C_F: ACTCAGTCTGGGTGGAAGGTATC, MYC_R: GGAGGAATGATAGAGG CATAAGGAG, AKNA_F: CCGTTCCAATCCCTTACC, AKNA_R: TGGAACAA AGAATTCACAGG, APRT_F: GCCTTGACTCGCACTTTTGT, APRT_R: TAG GCGCCATCGATTTTAAG, ChIA_neg_F: AGTCTGAGCTTTGTGGACAGC, and ChIA_neg_R: CCCTCCCAGTATACAGTCTTGC. qPCR, library preparation, and next-generation sequencing were performed as previously described[25]. Values of all qPCR replicates are supplied in the Source Data file.

**Western blot analysis**. Western blot analysis of equal amounts of crosslinked chromatin extracts or of histone extracts (see Supplementary Methods) was performed as described[25]. Antibodies used for western blot analysis are the following: anti-H3K27ac (ab4729, Abcam), anti-H3K27me3 (07-449, Millipore), 8WG16 (05-952, Millipore), anti-elongating RNA pol II (ab5095, Abcam), anti-Lamin B1 (ab65986, Abcam), anti-histone 4 (ab10158, Abcam), and anti-histone3 (ab1791, Abcam). Dilutions of antibodies were performed according to the manufacturer's guidelines. Time for analysis is indicated in the figures (Fig. 4b; Supplementary

Fig. 4a, b; Supplementary Fig. 3c, d). Uncropped scans of all western blot figures are supplied in the Source Data file.

**Assay for transposase-accessible chromatin (ATAC)-seq**. ATAC-seq method (nuclei preparation, transposition, and amplification of transposed fragments for library preparation) was performed using Nextera DNA Library Prep Kit (Illumina, Inc.) and primers as described by Corces et al.[39] with minor modifications: (i) 70,000 cells were used per experimental condition and (ii) the DNase treatment of cells in culture medium, before the transposition reaction, was skipped. The UV dose applied for ATAC-seq experiments was 15 J/m², and treated cells were left to recover for 2 h before harvesting.

**Start RNA isolation and qPCRs**. To isolate small RNAs (smaller than 200 nucleotides), we used Qiagen miRNeasy Mini Kit and RNeasy MinElute Cleanup Kit according to the manufacturer's instructions. In order to monitor the efficiency of the different enzymatic reactions, we included in our experiments a spike-in RNA oligonucleotide of known sequence (oGAB11: rArGrUrCrArCrUrUr-ArGrCrGrArUrGrUrArCrArCrUrGrArCrUrGrUrG, synthesized and purified by IDT). After purification, small RNAs and spike-in molecules were ligated to the IDT DNA linker 1 (/5rApp/CTGTAGGCACCATCAAT/3ddC/). Specifically, samples were denatured for 2 min at 80 °C and then placed immediately on ice. Ligation mix (4.8 µl of 50% PEG, 2 µl of 10× RNA ligase buffer, linker and RNase-free H₂O, and 0.5 µl of truncated RNA ligase) (NEB, Cat No. M0351S) was added in a final volume of 20 µl. The reaction was incubated for 3 h at 37 °C. After H₂O was added to a final volume of 100 µl, ethanol precipitation (three volumes of 100% EtOH, with 1/10th volume of 3M NaAc, pH 5.2, and 10 µg of glycogen (ThermoFisher Scientific, Cat No. AM9510) was performed overnight at −80 °C. RNA was purified at 10 µl, and reverse transcription (RT) was performed using primer oLSC003: /5Phos/TCGTATGCCGTCTTCTGCTTG/iSp18/CACTCA/iSp18/AAT-GATACGGCGACCACCGATCCGACGATCATTGATGGTGCCTACAG according to Invitrogen Superscript II (Cat No. 18064014) instructions. qPCR was performed using gene-specific forward primers (sequences 5′ to 3′ for OSBPL9: ATTGGCGGCTCCCAAGAT, SSBP1: GTGAGGGAGGAAGGGATAGC, IFIT1: TCTCAGAGGAGCCTGGCTAA, and KPNA6: ATTTGGCGAGAGCCTGTCT) and one common reverse primer (oNTI230: 5′-AATGATACGGCGACCACCGA-3′), which anneals to RT primer oLSC003 sequence. Quantitative PCR results were obtained from two independent biological experiments. Values of all qPCR replicates are supplied in the Source Data file.

**Read alignment, normalization, peak calling, and differential accessibility analysis**. For all next-generation sequencing (NGS) data analyses, in-house scripts and pipelines were developed to automate and analyze the data consistently (see below for details). Code is available upon request. Sequenced data and generated wig profiles are available on Gene Expression Omnibus (GEO) (Accession ID: GSE125181). ATAC-seq and ChIP-seq reads were subjected to quality control, data filtering, and alignment, and wig profile generation was performed essentially as described previously[25] with minor modifications. Chip-seq data for Pol II-ser2P and Pol II-hypo (Figs. 2 and 3; Supplementary Figs. 3, 6) were obtained from ref. [25] for NO UV and 2 or 1.5 h post UV (8 J/m²), respectively. nRNA-seq data were generated in our lab previously[25] (GSE83763) and obtained from ref. [27] (GSE75398). Downloaded data were processed as described in ref. [25] (see Fig. 5; Supplementary Fig. 5).

For H3K27ac and H3K27me3 ChIP-seq alignment files, peak calling was performed using SICER version 1.1[71] with window parameter = 400 bp and gap parameter = 1, while false discovery rate (fdr) and log₂ fold change cutoffs were set to 0.01 and 1.5, respectively.

For ATAC-seq alignment files, peak calling was performed using MACS2[72]. Because of the variability of ATAC-seq fragment lengths, several runs of the peak-calling algorithm were performed, using different parameters per run, in an attempt to maximize the sensitivity of the detection of open-chromatin regions. In particular, --nomodel --shift 100 --extsize 200, --broad --shift 100 --extsize 200, --nomodel --shift 37 --extsize 73, --broad --shift 37 --extsize 73, --broad --nomodel --shift 37 --extsize 73 --keep-dup all, --broad --nomodel --shift 100 --extsize 200 --keep-dup all and --nomodel --shift 75 --extsize 150 --keep-dup all runs were combined, and detected peaks were filtered using fdr < 0.05 and fold change > 1. Only peaks present in five out of seven methods were kept for further analysis. Although the majority of peaks detected were common between conditions, a number of peaks were also detected only in NO UV and +UV. As these peaks were less in number than the common ones, and showed a rather low density of reads in another attempt to maximize the sensitivity of the detection of open-chromatin regions, and to unbiasedly investigate the changes occurring upon irradiation, we considered the union of the peaks in each condition to perform the rest of the analysis (Supplementary Table 1). To conduct differential accessibility analysis, diffBind R package (https://www.bioconductor.org/packages//2.10/bioc/html/DiffBind.html) was used, with the merged ATAC-seq peak set as a reference. Differential accessibility regions were detected and filtered by applying fold change (Log₂ FC ≥ 1) and P-value (p-val ≤ 0.001) thresholds.

**Read-density plots**. ATAC-seq, ChIP-seq, nRNA-seq, CAGE-seq, and XR-seq data were subjected to read-density analysis after read depth normalization of all samples per experiment. Heatmaps and average-density profiling were computed as described previously[25] around genomic regions of interest, as indicated in the figures. Heatmaps were generated directly using the software, from matrices of binned read densities (bin size is indicated in the figures) for all considered individual (*n*) items (metagenes). Read-density matrices were also imported in R and python custom scripts for (i) plotting average-density profiles (smoothing achieved by a moving window of the bin size as indicated) and (ii) for determination of read densities per genomic category.

**Construction of mRNA-TSSs, PROMPT-TSSs, and eTSS annotation**. To annotate transcription start sites (TSSs), all known protein-coding and non-coding RNA hg19 RefSeq transcripts release 86 was downloaded from UCSC table browser (http://genome-euro.ucsc.edu/cgi-bin/hgTables). For each transcript, a biotype was assigned using BioMart (www.biomart.org), and all the small non-coding RNAs were excluded. For all the gene models containing multiple alternative transcripts, TSS neighborhoods of a 100-bp window were clustered together, and only the longest transcript was kept, resulting in 30,473 transcripts. Transcripts were then separated into three groups, based on their transcriptional activity. TSS coordinates were extended to 2 kb on each direction, and were tested for overlap with the Pol II-ser2P-UV, H3K27ac-UV, and H3K27me3-UV peak sets. Regions overlapping with at least one Pol II-ser2P-UV and H3K27ac -UV peak were characterized as active, those overlapping with a H3K27me3-UV peak, but not with a Pol II-ser2P-UV or with a H3K27ac-UV peak, were characterized as repressed, and those that did not overlap with any of the above peak sets were characterized as inactive. Any region overlapping with both H3K27ac-UV and H3K27me3-UV peaks was excluded from the rest of the analysis. This resulted in 15,819 active, 2943 repressed, and 7608 inactive transcripts (Supplementary Table 1). To further classify the active TSSs in terms of transcription directionality, the annotation was split up into unidirectional and bidirectional references. All active transcript pairs with opposite direction of transcription (forward(+) vs reverse(−)), where $-2\,kb \leq TSS_{distance} \leq +2\,kb$, $TSS_{distance} = TSS_{coordinate\_forward\,strand} - TSS_{coordinate\,reverse\,strand}$ (inter-TSS distance), were characterized as bidirectional, while the rest of the annotations were characterized as unidirectional. Bidirectional pairs were further categorized into two groups of annotations: convergent bidirectional transcript pairs with $TSS_{distance} \leq 100\,bp$, and divergent bidirectional transcript pairs with $TSS_{distance} > 100\,bp$. To optimize the categorization of convergent and divergent transcript pairs, TSS coordinates were redefined by scanning at a radius of 250 bp, to detect the nucleotide occupied by the maximum-sense CAGE signal. Any bidirectional pair with a non-significant CAGE peak in the aforementioned region was excluded from the analysis. This finally resulted in 12,859 unidirectional transcripts and 2822 active bidirectional TSS pairs, 1806 of which were characterized as divergent and 1016 as convergent (Supplementary Table 1).

To gain a complete overview of the non-coding antisense transcription events occurring around mRNA-TSSs, we also annotated upstream antisense (uaRNA) and downstream antisense (daRNA) transcripts (referred as an ensemble to PROMPTs in this paper for convenience). Only the active unidirectional mRNA-TSSs were used. For all the genes annotated with more than one mRNA transcript, only the leftmost TSS (for + strand genes), and rightmost TSS (for − strand genes) were considered for the rest of the analysis. The antisense CAGE peak with the highest summit in the region that ranged from −2 kb upstream to +1 kb downstream of each unidirectional TSS was considered to be the main PROMPT TSS for further analyses (inter-TSS distance = mRNA TSS - CAGE PROMPT). The above procedure was also repeated for the inactive transcript set to estimate the highest CAGE summit background distribution. The putative active PROMPT CAGE summits, which were higher than the average of the summit background distribution, were considered as active. This resulted in 5366 pairs of active unidirectional—PROMPT-TSSs, which were categorized to 1444 divergent and 3922 convergent pairs, as described above (Supplementary Table 1). By focusing on the divergent loci, the dynamics of transcription could be studied at play in each direction, without having to deal with interference from either direction. Therefore, analysis was focused on upstream antisense RNA, which corresponds to the original definition of PROMPTs[6].

To annotate enhancer transcription start sites (eTSSs), all 65,423 human enhancers from phases 1 and 2 of the FANTOM5 project from http://fantom.gsc.riken.jp/5/datafiles/phase2.2/extra/Enhancers/human_permissive_enhancers_phase_1_and_2.bed.gz, and the center of each annotation, were considered as the corresponding transcription start site. Enhancers were separated into 6766 active, 4730 repressed, and 39,227 inactive following the pipeline described above. Active intergenic enhancers were further analyzed, and all the eTSSs within a distance of 10 kb from nearby active transcripts, or neighbor eTSSs within a distance of 2 kb, were excluded. The rest of the intergenic eTSSs were extended to 1 kb in both directions, and sense and antisense maximum CAGE summit heights were detected for each reference. This procedure was also repeated for the inactive enhancer set, and inactive sense and antisense highest CAGE summit background distributions were estimated as described above. Finally, the putative active intergenic sense and antisense CAGE summits, which were higher averages of the summit background distributions, were considered as active. This resulted in 1228 active intergenic eTSSs (Supplementary Table 1).

**Promoter escape indices analysis**. Promoter escape analysis was performed for a subset of active unidirectional and bidirectional transcripts, PROMPTs and active enhancers. In particular, to avoid the inclusion of Pol II-ser2P reads mapped in overlapping promoters and gene bodies, only active divergent unidirectional transcript—PROMPT pairs were considered, where $TSS_{distance} > 100\,bp$, $TSS_{distance} = TSS_{coordinate\,forward\,reference} - TSS_{coordinate\,reverse\,reference}$, active divergent bidirectional transcript pairs with $TSS_{distance} > 100\,bp$, and active intergenic enhancers with no nearby transcripts within 10 kb and no nearby eTSSs within 2 kb. TSSs and PROMPT-TSS promoter escape indexes (EI, inverse of pausing index) were calculated as previously defined[25], by taking the average coverage in rpm (density in gene body was abbreviated as Db and ranged from 101 bp to 2 kb downstream of TSS or 101 bp downstream of TSS to TTS for genes larger or smaller than 2 kb, respectively) divided by the average coverage on the promoter-proximal region (Dp) ranged from 250 bp upstream to 100 bp downstream of TSS.

For enhancer escape analysis, EI was calculated as above, where density of reads at enhancer flanks (Df) is calculated for the regions ranging from −2 kb to −100 bp upstream of eTSS and from +100 bp to +2 kb downstream of eTSS, while density of reads on enhancer TSS (De) is calculated for the regions ranging from 100 bp upstream to 100 bp downstream of eTSS.

**Nucleotide excision repair data meta-analysis**. The strand-specific genome-wide maps of nucleotide excision repair of the UV-induced DNA damage (CPDs), available for XP-C mutants lacking the global genome nucleotide excision repair mechanism (GG-NER-deficient, TC-NER-proficient), were obtained from Hu et al.[20] (UV: 20 J/m², data used in Fig. 6, Gene Expression Omnibus (GEO) accession number GSE67941) and Chiou et al.[38] (UV: 20 J/m² data used in Fig. 7, GEO accession number GSE106823). XR-seq data for wild-type (WT) cells (UV: 10 J/m² used for Fig. 8 and Supplementary Fig. 7, GEO accession number GSE76391) were obtained from Adar et al.[46]. Sequence read archive (SRA) data sets were downloaded from Gene Expression Omnibus using the sra toolkit prefetch (https://www.ncbi.nlm.nih.gov/sra/docs/sradownload/) command, and converted to fastq files using fastq-dump. Fastq quality control, data filtering, and short read alignment were performed as above. Meta-analysis involved that read counts were normalized to equal read depth. Heatmap read-density matrices and average read-density plots of CPD XR-seq read around potential pyrimidine dimers. For XP-C data, we focus on TpT sequences (TT) as explained previously[25], and filtered the TTs overlapping with enhancer regions defined above to avoid signals generated from eTSSs. For WT cells, the list of TTs was further curated to only consider TTs in active genes (or inactive genes) located on plus strand and filtering out TTs in the region between TSS and +2 kb for CONV and DIV bidirectional and unidirectional genes with inter-TSS < 100 bp (Supplementary Table 1). XR-seq reads were computed as described above. Read-density matrices were calculated for both strands (TS and NTS) separately when indicated. For WT cells, only minus reads were used as these correspond to excised DNA from TS for plus genes. The ratio of XR-seq reads between directions, and calculation of variability between directions was performed as described in the legend of Fig. 6. "S–F" scores and quantification of reads around TT loci was performed as described in Lavigne et al.[25], with cluster borders defined previously.

**FANTOM5 Cap-analysis of gene expression (CAGE) sequencing data meta-analysis**. The FANTOM5 strand-specific CAGE-seq alignment files of normal dermal fibroblast primary cells (six donors with source codes: 11269-116G9, 11346-117G5, 11418-118F5, 11450-119A1, 11454-119A5, and 11458-119A9) and normal skin fibroblasts (two donors with source codes: 11553-120C5 and 11561-120D4) were downloaded from ftp://ftp.biosciencedbc.jp/archive/fantom5/datafiles/phase2.2/basic/human.primary_cell.hCAGE and were combined. Heatmap read-density matrices and average read-density plots were computed as described in the section "Read densities heatmaps and average plots". Read-density matrices were calculated for both strands separately.

**Reporting summary**. Further information on research design is available in the Nature Research Reporting Summary linked to this article.

## Data availability
The data reported in this paper have been deposited with the Gene Expression Omnibus under accession code GSE125181.

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

## Acknowledgements

We thank Pantelis Hatzis, Mihalis Verykokakis, and members of the Fousteri lab for critical discussions and reading of the paper. We thank Panagiotis Moulos for help and discussion on the underdevelopment computational pipelines. We thank Vladimir Benes and the Genecore facility (EMBL, Germany) for the special care they use in sequencing our NGS libraries. This work was funded by a European Research Council grant to M.F., Agreement-309612 (TransArrest) and <Matching Funds> to MF funded by National sources.

## Author contributions

M.F. and M.D.L. designed the study and were responsible for interpretation of the results. M.F. directed the study and obtained financial support. M.D.L., M.F., and A.L. wrote the paper and all authors edited the paper. A.L. performed the experimental part of the study. D.K. performed the statistical and bioinformatics analyses. M.D.L. contributed significantly in the bioinformatics analysis of the data. All authors discussed the results, reviewed, commented on, and approved the final version of the paper. A.L and D.K. contributed equally to the paper.

## Competing interests

The authors declare no competing interests.
