## [Peer Review File · Nature Communications]

Reviewers' comments:

Reviewer #1 (Remarks to the Author):

This paper is a continuation of the corresponding authors' previous work on the transcriptional response to UV damage in human cells. In their previous publication, they nicely showed that upon UV damage, prior to the shut-down of transcription, there is a coordinated release of paused RNA PolII from promoters, that could facilitate detection of DNA damages and repair by transcription-coupled repair.

This manuscript describes three novel findings.

- 1) That the promoter regions remain and even become more "open" after UV.
- 2) That there is no discernable change in either the activating histone mark H3K27ac or the repressive H3K27me3 mark after UV.
- 3) That after UV, there is a marked reduction in hypophosphorylated RNA PolII at transcription initiation sites, but that this is not due to inhibited Pol II hypo recruitment, but rather due to its more efficient release from the proximal promoter pause site.

In general, I find this manuscript to be very interesting, and can potentially add to mechanistic understanding of transcriptional responses and regulation in general, and the response to UV in particular.

Major comments:

- 1) Timeline. As the authors clearly know and point out, the response to UV is very dynamic. However, in the current manuscript – it is unclear when chromatin was assayed, specifically for the genomic ATAC- and histone ChIP- seq experiments. Western blots in supplementary figure 3 start from 3h after UV. However – at this stage – transcriptional response is well underway. Interpreting and evaluating their findings is very difficult without knowing when chromatin was assayed.
- 2) There have been several previous reports of hyper-acetylation in response to UV (in yeast and human systems) that contradict the authors findings for H3K27ac (examples include Wang J. et al Cancer research 2015, Ramanathan and Smerdon, Carcinogenesis 1986). The authors mention Schick S. et al J.Cell Sci. 2015 that saw major changes in H3K27ac after UV and explain the contrast by the different UV doses . However, once more, the dynamics of the processes could be playing a role here as well, and once more, not knowing the time at which chromatin was assayed makes it difficult to judge.
- 3) As the authors state in the introduction, it has been assumed in the field that transcription initiation shuts down after UV and is resumed after ~24h. It's not clear then, if PolII keeps being recruited, and released – are the authors saying transcription doesn't shut down at all – and the reduced levels of RNA synthesis are solely due to blocking damages?
- 4) I have a technical concern regarding the H3K27ac ChIP. Histone modification antibodies are not always as specific as companies claim they are. Consulting the histoneantibodies.com resource, it appears the abcam ab4729 antibody used in this study is one of the less-specific ones.

Minor comments:

- 1) In the analysis of ATAC-seq accessible regions:
 - a. I would like to see the analysis of the pre- and post-UV AR regions separately and not grouped together as in Sup. Figure 1a
 - b. I would like to see a confidence interval for the average plots, or the data for the two replicates separately.
- 2) In Figures 3-5:
 - a. the separation of the unidirectional genes by strand is very confusing. I would find it easier to unite the data, and analyze the transcribed strand of the major transcript on the right, and the

minor on the left.

b. Has the data been flipped in the negative strand genes in Figure 3? Because in 3b don't we expect the divergent transcripts to be on the top??

c. In 3 e. don't you expect the graph for negative genes to be in the opposite direction?

d. In 3g.h. there are two graphs for "genes" and it wasn't clear what the difference between them is (uni and bi-directional?)

3) In Figure 4b – PolIII Hyp westerns – I would expect there to be a negative (no UV, no DRB) control.

4) In Figure 7a: Could the 1h and 4h data for the template strand have been switched? The original Adar et al. manuscript described higher repair near TSS at 1h than at 4h...

5) Regarding the analysis in relation to XR-seq in DRB treated XP-C cells (End of p.11, beginning of p.12). This section is quite confusing and I had a hard time understanding it, and how they deduce that the data indicates there is concurrent Pol II recruitment and initiation.

6) In supplementary Figure 4d: The comparison we would like to make is between the +/- DRB. Separating them into two graphs makes that difficult.

Small technical comments

1) The Acronym CAGE is introduced prior to its specific name on page 6.

2) Defining doses and time points in the manuscript as "early" or "mild" is subjective. Please be specific.

In general, the manuscript is well written. The authors did a very thorough job of integrating their data with recent data from other publications, and relating to current theories in transcription regulation. If my comments are addressed, I think it will merit publication in Nature communications.

Reviewer #2 (Remarks to the Author):

This is mainly a ChIP-seq and meta analyses approach addressing the role of transcription dynamics in the repair of UV lesions. It is a continuation of the previously published work by the same group where they described a shift in the pol II distribution after UV stress (Lavigne et al., 2017, Nat Commun). In the previous work they showed that polymerases at promoters become released into productive elongation, which was conceptually new. The current study substantiates these findings. In addition, the authors present evidence that this release into productive elongation is accompanied by an increase in transcriptional initiation. The interesting new data are in Fig. 1 and Fig. 2, where the authors show by an ATAC-seq approach that UV radiation triggers a global increase in chromatin accessibility (Fig. 1), which occurs independently of changes in the H3K27ac and H3K27me3 levels but correlates with depletion of Pol II-hypo at transcriptional start sites (Fig. 2). Unfortunately, this correlation was not further investigated. It therefore remains unclear if the increased chromatin accessibility is the cause of the increased transcription initiation and elongation waves they observed previously. In addition, the mechanism of this increased chromatin accessibility remains unexplored. In summary, this is a solid paper and the data are of high quality and mostly support the conclusions drawn even though the conceptual advance is a bit limited (at least in comparison to their previous study published in this journal in 2017).

We appreciate the reviewers for their positive opinion on the relevance of our work. We believe that by taking into consideration their constructive comments the content and quality of the manuscript have greatly improved.

Please find below a point-by-point response to all the concerns raised by the reviewers. The Reviewers' comments are indicated in italics and are followed by our response in blue fonts. All changes made in the text are also highlighted in the revised manuscript.

Reviewer #1 (Remarks to the Author):

-

Reviewers' comments:

Reviewer #1 (Remarks to the Author):

This paper is a continuation of the corresponding authors' previous work on the transcriptional response to UV damage in human cells. In their previous publication, they nicely showed that upon UV damage, prior to the shut-down of transcription, there is a coordinated release of paused RNA PolII from promoters, that could facilitate detection of DNA damage and repair by transcription-coupled repair.

This manuscript describes three novel findings.

- 1) *That the promoter regions remain and even become more "open" after UV.*
- 2) *That there is no discernible change in either the activating histone mark H3K27ac or the repressive H3K27me3 mark after UV.*
- 3) *That after UV, there is a marked reduction in hypophosphorylated RNA PolII at transcription initiation sites, but that this is not due to inhibited Pol II hypo recruitment, but rather due to its more efficient release from the proximal promoter pause site.*

We thank the reviewer for his compliments regarding our previous work and also for highlighting the novel findings of our current study.

We wish to clarify that the observed increased accessibility is not limited to promoters, but to all active TSSs including those of functional regulatory regions expressing non-coding RNAs from PROMPTs and enhancers.

In general, I find this manuscript to be very interesting, and can potentially add to mechanistic understanding of transcriptional responses and regulation in general, and the response to UV in particular.

We thank the reviewer for acknowledging both the broad and more specific mechanistic insights of our study. Indeed, we consider the implications of our findings to be two-fold: 1) for the chromatin and transcription field our results provide a novel physiologically relevant evidence justifying why cells gain from continuously firing initiation from most of active genes in steady-state, although this process is often prematurely terminated at PPP. Also, mechanistically, our results reinforce the emerging concept that Pol II pausing has an inhibitory effect on the next molecule initiation, and show that PPP status determines the dynamics of initiation. 2) Our findings clarify long standing paradoxical observations in the field of UV-DDR and disclose how Pol II pause-release dynamics decrease the dwell time of the next recruited pre-initiating Pol II molecules that can continuously and readily initiate from essentially all active TSSs after genotoxic stress.

Major comments:

1) Timeline. As the authors clearly know and point out, the response to UV is very dynamic. However, in the current manuscript – it is unclear when chromatin was assayed, specifically for the genomic ATAC- and histone ChIP- seq experiments. Western blots in supplementary figure 3 start from 3h after UV. However – at this stage – transcriptional response is well underway. Interpreting and evaluating their findings is very difficult without knowing when chromatin was assayed.

The reviewer is right about the difficulty to find the relevant information on recovery times. We sincerely apologise about that. In the legends of Fig. 1, we forgot to point to the Methods section where conditions for the ATAC-seq are detailed. Actually, all the ATAC-seq and histone ChIP-seq experiments had been performed 2 hours post UV irradiation, a time-point that is considered within the early recovery phase (0-4 hours) and which we and others have defined as a critical time-point when the observed changes in the post-UV transcription response are maximal (i.e depletion of hypo-RNA Pol II). We have now indicated the irradiation dose and recovery time in the Figure legend for clarity.

Also, in the legends of Fig. 2, 3 and Fig. S6 for the histone modifications ChIP-seq experiments, we had not mentioned that the recovery time was 2h. We have now added this information.

To address the reviewer's comment and strengthen our findings we have now performed and included in the manuscript Western blot analysis of histone extracts performed at time-points post-UV covering the range of early recovery (+0.5h, +1h, +2h, +4h). This figure replaces the previously shown later time points for H3K27ac and H3K27me3 (old Supplementary Figure 3c, d) and characterizes how the dynamics of these two histone PTMs are limited during early recovery. In this way, one can make a direct comparison with the times used for our NGS analyses. We have updated the respective panels of Supplementary Figure 3c and 3d and the relevant text in the manuscript (page 5, line 154).

2) *There have been several previous reports of hyper-acetylation in response to UV (in yeast and human systems) that contradict the authors findings for H3K27ac (examples include Wang J. et al Cancer research 2015, Ramanathan and Smerdon, Carcinogenesis 1986). The authors mention Schick S. et al J.Cell Sci. 2015 that saw major changes in H3K27ac after UV and explain the contrast by the different UV doses . However, once more, the dynamics of the processes could be playing a role here as well, and once more, not knowing the time at which chromatin was assayed makes it difficult to judge.*

We thank the reviewer for his comment and once more, we apologise for not indicating in a more clear way the doses and times we had used. We have indeed discussed the possibility that when cells deal with unmanageable levels of damages they would need to implement different expression changes, required for the associated fate of programmed death. It has long been known that high UV doses induce apoptosis rather than repair of the damaged DNA (Li and Ho, *British Journal of Dermatology* 1998; Kulms & Schwarz, *Photodermatology photoimmunology & photomedicine* 2000; Farrell et al, *International Journal of Molecular Sciences* 2011). Cellular response is certainly different in these paradigms when cells are exposed to higher irradiation doses and this also leads to major phenotypic changes including chromatin compaction (Higuchi et al, *Annals of the New York Academy of Sciences* 2003; Ujvarosi et al, *Apoptosis* 2007). In line with these, Schick et al, *Journal of Cell Science* 2015 reported that 6 hours after UV irradiation (80 J/m²) there was a genome-wide drop in chromatin accessibility. They also reported a major re-organization of H3K27ac levels (also assayed at 6 hours post UV), suggesting that cells irradiated with high doses of UV get into the process of reprogramming and compaction of chromatin. This changes probably occur when the cells have sensed that the load of lesions is too high to be dealt with and they decide to favor a suicide path.

In contrast to the above, we chose in our study to focus on investigating how cells respond during the early recovery period after the induction of DNA lesions by a mild range of UV-C doses (8-20 J/m²). Our findings reinforce a model where chromatin relaxation has an active role in UV response (at least at early time points after irradiation). In fact, in addition to the increase in chromatin accessibility (as assayed by ATAC-seq 2 hours post UV (15 J/m²)), we observed a slight (though non statistically significant) increase of H3K27ac at TSS proximal regions (see Figure 2 and Supplementary Figure 3a). These findings are in accordance with the results of the Ramanathan and Smerdon, *Carcinogenesis* 1986 paper and also with the idea proposed by the other study mentioned by the reviewer; Wang et al, *Cancer Research* 2005.

We believe that our findings do not necessarily contradict the results of the above mentioned papers, especially given that these papers address either hyperacetylation of H4 or hyperacetylation of histones in general and not the levels and distribution of H3K27ac, as we do for the purposes of this manuscript. More specifically, Wang et al, 2005 describes how the increased hyperacetylation of H4 (but not of H3) that is observed in the first 5' to 1 hour in response to 200 J/m² depends on p33ING2, while, Ramanathan and Smerdon, 1986 paper shows that there is a small (10-30%) increase in core histone hyper- acetylation at early time points (2-4 hours) in response to different doses of UV (2 -20 J/m²). Notably, they found that the degree of global histone hyperacetylation decreases with increasing UV dose and approaches that of the control cells at about 12 J/m², a dose very similar to the one we have used (15 J/m²), thus pointing to similar trends with our findings for H3K27ac. We modified the manuscript accordingly (p. 14 line 352).

3) *As the authors state in the introduction, it has been assumed in the field that transcription initiation shuts down after UV and is resumed after ~24h. It's not clear then, if PolII keeps being recruited, and released – are the authors saying transcription doesn't shut down at all – and the reduced levels of RNA synthesis are solely due to blocking damages?*

Indeed, this is what we think is going on. The overall reduced levels of RNA synthesis derives from the slower transcription elongation rate that is observed after UV irradiation (Lavigne et al, *Nature Communications* 2017; Williamson et al, *Cell* 2017), which occurs due to the blockage of a proportion of the elongating Pol II at the UV lesions (see Fig. 4 in Lavigne et al, 2017). We have actually estimated in this manuscript that 63.65 % of the transcribed genome shows reduction in transcription activity (page 7). However, a long-lived (0-12h) local increase in nRNA synthesis downstream of TSS of all active genes is detected during the UV-recovery phase (Lavigne et al, *Nature Communications* 2017; Williamson et al, *Cell* 2017; Magnuson et al, *Scientific Reports* 2015; Andrade-Lima et al, *Nucleic Acid Research* 2015). This observation combined with the findings on the UV-induced chromatin opening and maintenance of activating histone marks are hardly compatible with the previously suggested model of UV-induced global transcription initiation shut down.

In the present paper, we propose an alternative model demonstrating that i) Pol II is continuously recruited to active TSSs after UV during the first 2-4 hours, when transcription initiation was assumed to be maximally inhibited and ii) Pol II productively synthesizes significant levels of start-RNA molecules, which can be detected between TSS and promoter proximal regions between 2 and 2.5 hours post-UV, thus clearly demonstrating an active transcription initiation process after UV. As we have also mentioned in the manuscript, these results show that the lower levels of Pol II-hypo detected by ChIP-seq upon UV does not reflect inhibition of Pol II-hypo recruitment, but rather shorter dwell time of this isoform in PIC at TSSs due to the more efficient transition into initiation, which is powered by the release of the downstream polymerase from the PPP sites into elongation upon UV.

4) *I have a technical concern regarding the H3K27ac ChIP. Histone modification antibodies are not always as specific as companies claim they are. Consulting the histoneantibodies.com resource, it appears the abcam ab4729 antibody used in this study is one of the less-specific ones.*

We thank the reviewer for pointing this out to us. We were surprised to see that, according to the ELISA-based database histoneantibodies.com, the Abcam ab4729 antibody we used for H3K27ac ChIP-seq is shown as one of the less-specific. Especially as the reason we chose this particular antibody was because that was one of the most widely used in research (865 references to date according to manufacturer's website). Moreover, by searching some of these references we found that many recent high quality papers use this antibody for H3K27ac ChIP-seq (Liu X et al., *Nature* 567:525-529 (2019); Pastore A et al. *Nature Communications* 10:1874 (2019); *Nature Communications* 10:1054 (2019)). Notably, this antibody has also been used by the Roadmap Epigenomics Project (<http://www.roadmapepigenomics.org>) in ChIP-seq experiments to assess the genome wide binding profile of H3K27ac in cells of the different human tissues (e.g. <https://www.ncbi.nlm.nih.gov/geo/query/acc.cgi?acc=GSM958163>).

However, as we were challenged and intrigued by the reviewer's comment we looked for H3K27ac ChIP-seq data generated with the higher ranking antibody tested by histoneantibodies.com: the Active motif antibody 39133. We found and analysed ChIP-seq data generated from primary skin fibroblast cells (cell line GM23248 (https://www.coriell.org/0/Sections/Search/Sample_Detail.aspx?Ref=GM23248&product=CC), available from the ENCODE project (accession number: ENCSR007YOT). We compared the

ENCODE anti-H3K27ac data generated using the active motif Antibody to our ChIP-seq data of NO UV condition (steady state condition) generated in the VH10 hTERT immortalized foreskin fibroblast cells (Kolman et al., *Environmental & Molecular Mutagenesis*, 1992). Although these two data sets have not been generated in the same cells, our results (see Response Figure 1, below) show a good correlation (PCC=0.76) of the location of peaks and the underlying density of reads that were detected with either antibody fully supporting our belief that the reported distributions are reliable. We also note that the peptide arrays database histoneantibodies.com might not be the most suitable tool to assess the specificity of histone PTM antibodies in non-denaturing applications such as ChIP-seq, where the modification is recognized in a more physiologic state. In agreement, the Authors of histoneantibodies.com (Rothbart et al, *Molecular Cell* 2015) reported *<the arrays may not represent assays like ChIP in that ...the nucleosomes ...remain largely intact and folded.>*

Response Figure 1: Comparison of H3K27ac ChIP-seq performed with Abcam (ab4729) and Active motif (39133) antibodies. **a.** Heatmap (upper) and average plot (lower) depicting H3K27ac ChIP-seq read densities around peaks detected with one or the other antibody in VH10 or GM23248 cells as indicated. Genomic regions 5 kb around peak centers that overlap with active TSS and eTSS are depicted. **b.** Correlation scatter plot showing normalized read counts for H3K27ac ChIP-seq in GM23248 (Active motif antibody) and VH10 cells (Abcam, antibody). Pearson Correlation Coefficient (PCC) was calculated and reported on the graph. **c.** UCSC snapshot depicting H3K27ac ChIP-seq signal for GM23248, VH10 NO UV and VH10 +UV (2h) with antibodies as indicated.

To check if our antibody performs well in denaturing conditions (SDS-PAGE followed by Western blot) we did a new series of experiments for histone extracts (in triplicate) assessing the effect of UV on histones PTMs at earlier times (see response to comment 1 above). When incubating different blots with either the H3K27ac antibody ab 4729 from Abcam (New Supplementary Fig. 3c and Response Figure 2) or with the Active motif 39133 antibody (Response Figure 2) we observed very similar patterns, demonstrating that irradiation of cells with mild UV doses (15 J/m²) does not have a

significant change on the bulk levels of H3K27ac during the first 0.5 to 4 hours after UV, in line with initial findings.

Response Figure 2. Western Blot analysis of histone extracts for bulk H3K27ac levels. (Left panel) The antibody used for H3K27ac detection was the Active Motif 3913. (Right panel) The antibody used for H3K27ac detection was the abcam 4729. UV dose was 15 J/m². Each blot is representative of three independent biological experiments, and quantification of the variability across time points suggest no difference between the two antibodies (Ordinary one-way ANOVA Dunnett’s multiple comparisons test non-significant (n.s) between NO UV and +UV conditions).

Taken together our data demonstrate that both ab 4729 and Active motif antibody can be confidently used in ChIP-seq and Western blot analysis. We therefore conclude that our H3K27ac analysis is acceptable in both specificity and quantitative manner to draw reasonable biological conclusions.

Minor comments:

- 1) *In the analysis of ATAC-seq accessible regions:*
 - a. *I would like to see the analysis of the pre- and post-UV AR regions separately and not grouped together as in Sup. Figure 1a*

We depict for the reviewer here the analysis of ATAC-seq reads on the pre- and post-UV AR regions separately (Response Figure 3). Although the majority of peaks detected are common between conditions (63981), we detected 2012 peaks called only in NO UV and 40089 peaks called only in +UV (Response Figure 3e). We noticed that the peaks being unique for each experimental condition are lower in signal than the common ones and a small number of reads (under peak calling threshold) is also present in the other condition. For this reason and to avoid biasing our analysis towards the most accessible fraction of the genome, we have unified the regions detected by peak calling for the purposes of this manuscript. In this way, we have a more global and representative views of what are the possible changes in chromatin accessibility after UV.

We added this sentence in the Materials and Methods section (p. 21, line 569) to be clearer:

” Although the majority of peaks detected were common between conditions, a number of peaks were also detected only in NO UV and +UV. As these peaks were less in number than the common ones

and showed a rather low density of reads in another attempt to maximize the sensitivity of the detection of open chromatin regions and to unbiasedly investigate the changes occurring upon irradiation, we considered the union of the peaks in each condition to perform the rest of the analysis.”

Response Figure 3. Analysis of the pre- and post-UV AR regions separately a. (Left) Heatmap depicting read densities of ATAC-seq NO UV peaks (n=65990) in NO UV and +UV (+2h) condition at promoter, intragenic and intergenic regions, respectively. **b.** Venn diagram illustrating the proportions of only NO UV, common, and only +UV peaks for promoter, intragenic and intergenic regions. **c.** (Left) As in **a** but for ATAC-seq +UV peaks (n=104067). **d.** Distribution of NO UV (blue border) and +UV (pink border) ATAC-seq peaks in promoter (+/- 2 kb relative to TSS), intragenic and intergenic regions. **e.** (Top) Venn diagram depicting the overlap of all peak regions in NO UV and +UV conditions. (Bottom) Heatmaps (arranged by k-means clustering) and average profiles

depicting read densities in NO UV and +UV condition 1 kb around the center of peaks separated in only NO UV (left), common (middle) and only +UV (right) peaks.

b. I would like to see a confidence interval for the average plots, or the data for the two replicates separately.

We have now included a heatmap of the separate ATAC-seq replicates in Supplementary Fig. 1b. We also show in the Response Figure 4b how the average profiles for each replicate plotted around all ATAC-seq peaks. The quantification of reads coming either from replicate 1 or replicate 2 for NO UV (-UV) and + UV show that one can be confident that replicates are reproducible and that overall there is a significant (c, 95% CI range of the difference between + UV and NO UV does not include 0) increase in ATAC-seq signal after UV irradiation at those regions.

Response Figure 4. a. Heatmap depicting ATAC-seq peaks of two biological replicates separately (Rep 1, Rep 2) for NO UV and +UV condition in promoter, intragenic and intergenic regions as described if Fig. 1b for the union of peaks detected in NO UV and + UV conditions. Peaks are sorted by increasing read density. **b.** Average plots depicting signal of ATAC-seq peaks for each biological replicates for NO UV and +UV conditions. **c.** Box plot depicting the distribution of reads densities around ATAC peak centers (-500; +500 bp) when considering both Rep 1 and Rep 2 for NO UV and + UV. Box plots show the 25th–75th percentiles, and error bars depict data range to the larger/ smaller value no further than 1.5 * IQR (inter-quartile range, or distance between the first and third quartiles). 95 % confidence intervals (CI) of mean differences between + UV and NO UV read counts were calculated for 10,000 samplings of 100 data points with replacement from each population. Difference is considered significant if range does not include 0.

2) In Figures 3-5:

a. *the separation of the unidirectional genes by strand is very confusing. I would find it easier to unite the data, and analyze the transcribed strand of the major transcript on the right, and the minor on the left.*

We are sorry that the way we organised the data was confusing. We, in fact separated unidirectional genes encoded on plus or minus strands of the genome to specifically show in any case what is happening on major (mRNA) and minor (PROMT) transcript encoding regions. As depicted in the schematic on the left (Figures 3-5) for mRNA genes encoded on minus strand we inverted the image (the - strand is on the top). In this way, as the reviewer suggested, we show the transcribed strand of the major transcript on the right, and the minor on the left in any instance. In this way, it is simpler to observe and interpret what is going on for the major transcript (always displayed on the right of TSS) and the minor transcript (always displayed on the left of TSS). We now added labels on the figures for clarity to explain that the upper panel represents plus genes and the lower panel minus genes.

b. *Has the data been flipped in the negative strand genes in Figure 3? Because in 3b don't we expect the divergent transcripts to be on the top??*

Please see our previous reply. The divergent transcripts to mRNA genes (whether they are encoded on minus or plus strand) are displayed towards the bottom (labelled as DIV, with increasing distance calculated between mRNA's TSS and PROMPT's TSS).

c. *In 3 e. don't you expect the graph for negative genes to be in the opposite direction?*

Based on our explanation above, it is expected to see on the average plot the minor transcripts on the left (upstream) of TSS and the major transcript on the right (downstream) of TSS. We believe that it is easier this way to see that all unidirectional genes show the same pattern.

d. *In 3g.h. there are two graphs for "genes" and it wasn't clear what the difference between them is (uni and bi-directional?)*

Yes, the reviewer is right, although we noted in the legend that the 'genes' refer to bidirectional in one graph and unidirectional in the other, it wasn't clearly labeled in the figure. In panel g we show (and have now labelled more clearly) EI analysis for bidirectional genes, and in h EI analysis for unidirectional genes (this includes both minus and plus mRNA genes).

3) *In Figure 4b – PolII Hyp westerns – I would expect there to be a negative (no UV, no DRB) control.*

We agree with the reviewer, however, in Supplementary Figure 4b we showed the untreated (No UV, No DRB) control for Pol II hypo Western Blot. The comparison of DMSO treated cells presented in this figure recapitulates our and previously published observations (Rockx et al, *PNAS* 2000, Heine et al, *Journal of Biological Chemistry* 2008, Lavigne et al, *Nature Communications* 2017) verifying that Pol II-hypo levels drop significantly upon UV. As we already presented this control in Supplementary Figure 4b, to avoid repetition, in the western blot shown in Figure 4b we analysed only the four conditions as depicted.

4) In Figure 7a: Could the 1h and 4h data for the template strand have been switched? The original Adar et al. manuscript described higher repair near TSS at 1h than at 4h...

We thank the reviewer for this comment. It certainly triggered a troubleshooting round that made us evaluate what causes these differences.

To verify that no mistake happened during the download and analysis of (Adar et al, *PNAS* 2016) CPD XR-seq data, we generated an average plot of the reads detected on the Template Strand (TS) of active genes from TSS to 10 kb and found a similar pattern to the one presented by Adar et al. (compare Response Figure 5 with Figure 1D in Adar et al, *PNAS* 2016), indicating that excision might be higher at 1h compared to 4h.

However, we also note that when the authors in the original article focused on what would be CPD-specific NER events employing the anti-CPD pull-down only (thus skipping the pulling-down with anti-TFIIH first) and analysed the amount of excised DNA fragments (see Fig. 1 A-B in Adar et al, *PNAS* 2016), it is then obvious that their data also demonstrate that the level of CPD-containing excised fragments is higher at 4 h than at 1 hour.

We thus believe that aligning the reads on TS TTs might enrich even more the TC-NER-specific events detected by XR-seq. As XR-seq involves an initial immunoprecipitation step by the transcription factor TFIIH, among the reads obtained with this protocol some might represent TSS-specific non-damaged DNA, which is filtered out when the reads are aligned to TTs.

CPD-XR-seq reads (Adar et al., 2016) on TS of active genes larger than 10 kb

Response Figure 5. Heatmap depicting reads density of CPD XR-seq signal on transcribed strand (TS) of NHF1 normal human fibroblasts at 1h, 4h and 8h after UV irradiation. Genomic regions 10kb downstream of TSS are depicted. Data obtained from Adar et al, 2016.

Looking at average profiles over gene bodies gave us the idea to look if we could also monitor the wave of excision going through the genes as time passes, as an alternative representation of what we showed for the TTs analysis that we performed only on reads aligning to TS (Fig. 7a-c, clusters III-V and VI). We now show (Supplementary Fig. 8) that potentially damaged TT loci located more distally

to TSSs on long active genes (from 50 kb to 100 kb) still get recognized and excised more efficiently as time passes by the TC-NER machinery (in comparison to GG-NER-inactive genes), as the continuously initiating polymerase can reach further in the gene bodies. We added the corresponding paragraph in the manuscript:

“We found that significant levels of transcription-dependent excision activity was maintained at lesions located directly downstream of active genes TSSs, especially at 4 h into the recovery process (compare Fig. 7 and Supplementary Fig. 8a-c). Notably, it also appeared that the bulk of excision activity on TS shifts overtime (+ 8 h) from the proximal to the distal part of long genes. Overall, these results reveal that a large extent of the transcription-driven repair activity detected after UV is due to the ongoing entry of Pol II molecules at TSSs, which can scan farther and farther lesions along the gene bodies.” (p. 12, line 322)

5) *Regarding the analysis in relation to XR-seq in DRB treated XP-C cells (End of p.11, beginning of p.12). This section is quite confusing and I had a hard time understanding it, and how they deduce that the data indicates there is concurrent Pol II recruitment and initiation.*

We are sorry that the reviewer found this part confusing. We now have tried to explain it in a more clear way. DRB and DRB2 experiments have been designed by Chiou et al, *Journal of Biological Chemistry* 2017 and are very insightful. The idea was to limit the number of Pol II molecules that enter the gene bodies of transcribed genes either before or immediately after UV and assess the effects on CPD excision repair. We have now updated the manuscript for clarity (p. 12, lines 300-319).

6) *In supplementary Figure 4d: The comparison we would like to make is between the +/- DRB. Separating them into two graphs makes that difficult.*

In the right panel of Supplementary Fig. 4d, we did actually present the effect of DRB in response to UV in a common graph. In particular, we depicted the average ChIP-qPCR enrichment of all promoters tested of both DMSO and DRB treated samples, revealing how applying DRB during the early-post UV- recovery rescues the loss of Pol II hypo at TSSs. In the left panel, we chose to show the actual values of the two conditions (DMSO +/- UV and DRB +/- UV) separately, as in the no UV condition inhibition of pause-release increases accumulation of Pol II hypo in chromatin of promoter proximal regions as has been reported previously (Marshall et al, *Journal of Biological Chemistry* 1996, Heine et al, *Journal of Biological Chemistry* 2008; Rahl et al, *Cell* 2010; Henriques et al, *Molecular Cell* 2013). This can be appreciated by comparing the fold enrichment of ChIP qPCR experiments for Pol II hypo (Left panel, please compare NO UV condition for DMSO and DRB treatments).

Small technical comments

1) *The Acronym CAGE is introduced prior to its specific name on page 6.*

We correctly introduced this acronym now (p. 6, line 163).

2) *Defining doses and time points in the manuscript as “early” or “mild” is subjective. Please be specific.*

To clarify these terms, we have now specified the timing and the UV dose of our experiments in the manuscript and clearly defined “early” and “mild” in term of recovery times and doses range. Please see highlighted text in the manuscript (see p. 3 line 98 and p. 14 line 352).

In general, the manuscript is well written. The authors did a very thorough job of integrating their data with recent data from other publications, and relating to current theories in transcription regulation. If my comments are addressed, I think it will merit publication in Nature communications.

We thank the reviewer for his good words and his constructive comments.

Reviewer #2 (Remarks to the Author):

This is mainly a ChIP-seq and meta analyses approach addressing the role of transcription dynamics in the repair of UV lesions.

Indeed, the combination of next-generation sequencing approaches with an integrative analysis of ATAC-seq, ChIP-seq, nRNA-seq and XR-seq with/or without the use of inhibitors gave us the opportunity to precisely and functionally dissect the transcriptional response to UV stress.

It is a continuation of the previously published work by the same group where they described a shift in the pol II distribution after UV stress (Lavigne et al., 2017, Nat Commun). In the previous work they showed that polymerases at promoters become released into productive elongation, which was conceptually new. The current study substantiates these findings. In addition, the authors present evidence that this release into productive elongation is accompanied by an increase in transcriptional initiation.

We thank the reviewer and appreciate that he recognizes the novelty and importance of our studies. We would like to add that the current manuscript focuses on explaining how a previously misapprehended loss in Pol II-hypo at promoters is in fact a perfectly explainable evidence for the novel concept we put forward in this manuscript that continuation (not increase) in transcription initiation is a major player in determining the efficiency and homogeneity of transcription-blocking lesion repair in the transcriptome after UV.

The interesting new data are in Fig. 1 and Fig. 2, where the authors show by an ATAC-seq approach that UV radiation triggers a global increase in chromatin accessibility (Fig. 1), which occurs independently of changes in the H3K27ac and H3K27me3 levels but correlates with depletion of Pol II-hypo at transcriptional start sites (Fig. 2). Unfortunately, this correlation was not further investigated. It therefore remains unclear if the increased chromatin accessibility is the cause of the increased transcription initiation and elongation waves they observed previously.

We also believe that Fig.1 and Fig. 2 are important to show a novel aspect of UV transcriptional response. As we have discussed in the manuscript, there are previous reports demonstrating that rapid transcriptional induction can be accompanied by an increase in chromatin accessibility without changes in nucleosome occupancy (See page 13, paragraph 2 of Discussion; Mueller et al, *Genes and Development* 2017) and that higher levels of gene expression are associated with surges in chromatin

accessibility (Gray et al, *eLife* 2017, Ucar et al, *Journal of Experimental Medicine* 2017). We are able to see this in our data, as in steady-state there is a clear functional relationship between higher chromatin accessibility and increased transcription; more Pol II-ser2P and more nRNA in gene bodies are generally observed for genes with more “open” promoters (Response Figure 6).

In case of the UV-response, perhaps the reviewer did not notice that we show in Fig. 3 how the rapid release of Pol II from promoter-proximal pause sites (increase in EI) in active transcripts including PROMPTs and eRNAs is paralleled by chromatin relaxation around TSSs. Moreover, we present data indicating that this phenomenon is connected to the loss of Pol-II from TSSs (Supplementary Fig. 6).

We have now clarified the manuscript on this point (See Discussion p. 15, line 386):

“we propose that the rapid exchange of Pol II isoforms after UV irradiation represents a perfectly plausible cause for the decreased ability to detect Pol II-hypo molecules at TSSs upon UV. It also accounts for increase in EI, and could explain the gain in accessibility around TSS (see Fig.2-4, Supplementary Fig. 6).”

Response Figure 6. (Upper) Heatmap depicting read densities of ATAC-seq, nRNA-seq, ChIP-seq Pol II ser2P of control (NO UV) VH10 cells, for genes larger than 10kb, sorted by Pol II-ser2P density. Genomic regions 1kb upstream and 10kb downstream of TSS are depicted. (Lower) Same as in upper panel but for genomic regions 1kb around ATAC-seq peak centers. Peaks are sorted by increasing ATAC-seq read density.

In addition, the mechanism of this increased chromatin accessibility remains unexplored.

Indeed, this is a very good point. We are very interested to understand what could be the exact mechanism that governs this molecular response and we are aiming to address this question in a separate manuscript where we try to understand the changes in accessibility upon UV and the possible role of various chromatin regulators.

In summary, this is a solid paper and the data are of high quality and mostly support the conclusions drawn even though the conceptual advance is a bit limited (at least in comparison to their previous study published in this journal in 2017).

We thank the reviewer and appreciate his acknowledgement. We hope to have convinced him (with the revised version of the manuscript) that revealing the importance of continuation of transcription initiation and characterizing the underlying molecular events, is a game changer in understanding how cells efficiently and homogeneously drive the repair of transcription-blocking lesions through the whole transcriptome.

REVIEWERS' COMMENTS:

Reviewer #1 (Remarks to the Author):

I am generally very satisfied with the response to my comments. After clarification, it is my opinion that supplemental Figure 7 is indeed very informative and gives strong support to the authors model. I suggest it be moved into the main text.

Though the authors have put much effort into addressing it, I am not completely convinced regarding the authors response to my comment #4: " We thus believe that aligning the reads on TS TTs might enrich even more the TC-NER-specific events detected by XR-seq. As XR-seq involves an initial immunoprecipitation step by the transcription factor TFIIH, among the reads obtained with this protocol some might represent TSS-specific non-damaged DNA, which is filtered out when the reads are aligned to TTs."

If you examine the supplemental material to Adar et al. it appears that the frequency of TT is similar at 1h and 4h. Since there is also a CPD-IP in the XR-seq protocol, I am doubtful the differences in XR-seq between the time points is due to non-damaged DNA. However, it could be that the difference is explained by different repair kinetics/distributions of the other pyrimidine dimers (CT/CC/TC). Regardless, the results clearly show a difference at 8h compared to 1/4h and if the authors are convinced their data was not switched at the last phases of plotting - I accept their point and the figure as it is.

In summary, I recommend to accept the paper and look forward to the authors future work.

Reviewer #2 (Remarks to the Author):

I appreciate the care and thoroughness by which the authors addressed my comments and the comments of the other reviewer to their manuscript. I have no further points and recommend publication of this manuscript.

Reviewers' comments:

Reviewer #1 (Remarks to the Author):

I am generally very satisfied with the response to my comments. After clarification, it is my opinion that supplemental Figure 7 is indeed very informative and gives strong support to the authors model. I suggest it be moved into the main text.

We thank Reviewer 1 for his/her comments regarding Supplemental Figure 7. We have now moved the figure into the main text, according to his/her suggestion and labelled it as Fig. 7. The remaining figures and Supplementary Figures were also relabelled accordingly.

Though the authors have put much effort into addressing it, I am not completely convinced regarding the authors response to my comment #4: “ We thus believe that aligning the reads on TS TTs might enrich even more the TC-NER-specific events detected by XR-seq. As XR-seq involves an initial immunoprecipitation step by the transcription factor TFIIH, among the reads obtained with this protocol some might represent TSS-specific non-damaged DNA, which is filtered out when the reads are aligned to TTs.”

If you examine the supplemental material to Adar et al. it appears that the frequency of TT is similar at 1h and 4h. Since there is also a CPD-IP in the XR-seq protocol, I am doubtful the differences in XR-seq between the time points is due to non-damaged DNA. However, it could be that the difference is explained by different repair kinetics/distributions of the other pyrimidine dimers (CT/CC/TC). Regardless, the results clearly show a difference at 8h compared to 1/4h and if the authors are convinced their data was not switched at the last phases of plotting - I accept their point and the figure as it is.

The reviewer has a point here. As shown in the previous response round and in the former Fig. 7 (now Fig. 8) and former Supplementary Figure 8d (now Supplementary Figure 7d), we are convinced that the datasets of Adar et al. were not switched at the last phases of plotting. The average plots along gene bodies (TSS to +100 kb) we generated do show similar pattern with the ones originally published (TSS to 10 kb).

Also, we agree that when looking at the supplemental material in Adar et al. the frequency of TT is similar at 1h and 4h, when considering all possible dipyrimidine dimers. It is therefore possible that changes in relative XR-seq signal on the TTs reference vs gene body reference derive from the different repair kinetics/distributions of the other pyrimidine dimers not included in our reference (i.e. CT/CC/TC).

Nevertheless, as the reviewer correctly mentioned, the point we make in the manuscript underlines that a significant level of transcription-dependent excision activity is maintained at lesions located directly downstream of active genes TSSs even at 4 h and 8 h into the recovery process (compare new Fig. 8 and Supplementary Fig. 7a-c). We also highlight the fact that the bulk of excision activity on TS shifts overtime (from +1 h to + 8 h) from the proximal to the distal part of long genes (Fig. 8a-c, clusters III-VI, and Supplementary Fig. 7d).

In summary, I recommend to accept the paper and look forward to the authors future work.

We thank Reviewer 1 for his/her positive words and contribution in this productive review process.

Reviewer #2 (Remarks to the Author):

I appreciate the care and thoroughness by which the authors addressed my comments and the comments of the other reviewer to their manuscript. I have no further points and recommend publication of this manuscript.

We thank Reviewer 2 for his/her overall comments and his/her good words regarding our work.